# Softmax for Continuous Actions: Optimality, MCMC Sampling, and Actor-Free Control

## Abstract

As a mathematical solution to entropy-regularized reinforcement learning, softmax policies play important roles in facilitating exploration and policy multimodality. However, the use of softmax has mainly been restricted to discrete action spaces, and significant challenges exist, both theoretically and empirically, in extending its use to continuous actions: Theoretically, it remains unclear how continuous softmax approximates hard max as temperature decreases, which existing discrete analyses cannot handle. Empirically, using a stand actor architecture (e.g., with Gaussian noise) to approximate softmax is subject to the limited expressivity, while leveraging complex generative models can involve more sophisticated loss design. Our work address these challenges with a simple Deep Decoupled Softmax Q-Learning (DDSQ) algorithm and associated analyses, where we directly implement a continuous softmax of the critic without using a separate actor, eliminating the bias due to actor's expressivity constraint. Theoretically, we provide theoretical guarantees on the suboptimality of continuous softmax based on a novel volume-growth characterization of the level sets in action spaces. Algorithmically, we establish a critic-only training framework that samples from softmax via state-dependent Langevin dynamics. Experiments on MuJoCo benchmarks demonstrate strong performance with balanced training cost.

## 1 Introduction

Maximum entropy regularization is a standard framework in reinforcement learning to enhance policy multimodality (Ziebart et al., 2008; Haarnoja et al., 2017) and improve training robustness (Eysenbach & Levine, 2022), which induces the softmax distribution as a closed-form solution. Existing research on softmax (Song et al., 2019; Smirnova & Dohmatob, 2020) are often restricted to finite and discrete actions. When it comes to continuous actions (Van Hasselt & Wiering, 2007), a standard approach is to use a separate actor to optimize the entropy-regularized objective, which inevitably results in a discrepancy between the actor distribution and the real softmax policy (see Figure 4), especially when actor architecture induces relatively simple action distributions (Agarwal et al., 2021). While one can deploy more sophisticated generative architectures like diffusion or flow models for actor parameterization (Tang & Agrawal, 2020; Janner et al., 2022; Ma et al., 2025), instabilities can occasionally occur during training (Barceló et al., 2024), and the computational cost of the forward & backward processes can be high (Kang et al., 2023). On top of that, maintaining and tuning two separate complex neural-nets (the actor and the critic) at the same time can be demanding, and the complex losses handcrafted for generative models further adds to the complexity of the methods (Ajay et al., 2022; Black et al., 2023).

In this work, we propose a simple mitigation to the above problems, where we directly extend softmax to continuous actions and propose a critic-only algorithm without actors. In contrast to existing approaches that treat softmax policy approximation as an *optimization* problem for minimizing the distance between actor action distribution and the real solution to entropy regularization, we frame it as a *sampling* problem (Levine, 2018) and leverage Markov Chain Monte Carlo (MCMC) techniques, especially Langevin dynamics, to tackle sparsity and heterogeneity of high-dimensional spaces (Neal et al., 2011; Welling & Teh, 2011). The result is a simple, actor-free framework for deep continuous control, which we call Deep Decoupled Softmax Q-Learning (DDSQ). Our contributions can be outlined as follows.

- **Continuous Softmax Analysis.** In Section 3, we analyze the approximation error of softmax policies w.r.t. hard max. A suboptimality guarantee is established, based on a novel volume-growth characterization of the level sets in action spaces. We apply this result to provide convergence analyses of softmax value iteration for continuous actions.

- **Non-Parametric MCMC.** In Section 4, we formulate continuous softmax policies as a non-parametric, actor-free Langevin sampler. To promote more stable sampling, we propose several design choices, including SNIS initialization as an informative accelerator, specular reflection to handle boundary stagnation, and a careful selection of candidate step schedules.

- **Empirical Validation.** In Section 5, We train DDSQ across eight continuous control tasks in the MuJoCo suite. The main results demonstrate that DDSQ achieves strong performance with reasonable training time, while additional studies confirm its ability to capture multimodal policies and offer more flexible temperature control.

**Broader Relevance** Our work also bears significance beyond the specific scope of the paper, as softmax policies are an important tool that plays fundamental roles in RL theory, yet their use is mostly restricted to finite and discrete action spaces. In particular, softmax policies are the analytical solution to the entropy-regularized objective (Neu et al., 2017; Haarnoja et al., 2018), which has gained popularity in RLHF for LLMs recently (Christiano et al., 2017; Xiong et al., 2023; Chen et al., 2024). The use of softmax policies in RL can often be viewed as an application of *mirror descent* (Beck & Teboulle, 2003) and *natural policy gradient* (Kakade, 2001), which are frequently the key to achieving strong theoretical guarantees in both offline (Xie et al., 2023) and online RL (Liu et al., 2023). Our work, both the theoretical analyses in Section 3 and 4, and the practical implementation in Section 5, lays the foundation of extending the theoretical results in the literature to the more challenging continuous-action domains.

## 2 PRELIMINARIES

**Markov Decision Processes.** The Markov Decision Process (MDP) is represented by a tuple $(\mathcal{S}, \mathcal{A}, P, R, \gamma)$, where $P : \mathcal{S} \times \mathcal{A} \to \Delta(\mathcal{S})$ governs state transitions and $R : \mathcal{S} \times \mathcal{A} \to [0, R_{\max}]$ assigns scalar rewards. The central objective in policy optimization is to maximize the discounted return $\max_\pi J(\pi) := \mathbb{E}\left[\sum_{t=0}^\infty \gamma^t R(s_t, a_t)\right]$, where $s_0 \sim d_0$, $a_t \sim \pi(\cdot|s_t)$, and $s_{t+1} \sim P(\cdot|s_t, a_t)$. We define the *state-action value function* as

$$Q^\pi(s, a) = \mathbb{E}_\pi\left[\sum_{t=0}^{+\infty} \gamma^t R(s_t, a_t) \,\middle|\, s_0 = s, a_0 = a\right] \in [0, V_{\max}],$$

where $V_{\max} = \frac{R_{\max}}{1-\gamma}$. It represents the expected cumulative return starting from state-action pair $(s, a)$ under policy $\pi$. It is also the unique fixed point of the (policy-specific) *Bellman operator* $\mathcal{T}^\pi$, defined as $(\mathcal{T}^\pi f)(s, a) = R(s, a) + \gamma \mathbb{E}_{s' \sim P(\cdot|s,a)}[f(s', \pi)]$, where $f(s', \pi) = \mathbb{E}_{a' \sim \pi(\cdot|s')}[f(s', a')]$. The value function w.r.t. an optimal policy particularly serves as a unique solution to $f(s, a) = R(s, a) + \gamma \mathbb{E}_{s' \sim P(\cdot|s,a)}[\max_{a' \in \mathcal{A}} f(s', a')]$, the Bellman optimality equation.

**Softmax Policies and Softmax Value Iteration.** Given a measurable space $(\mathcal{X}, \mu)$ equipped with a base measure $\mu$ (the counting measure # in the discrete case and the Lebesgue measure $\mathscr{L}$ in the continuous case), an *entropy-regularized optimization* problem (Neu et al., 2017) aims to find a density $\pi$ w.r.t. $\mu$ that maximizes

$$\int_\mathcal{X} f(x)\pi(x)d\mu(x) + \lambda\mathcal{H}(\pi), \quad \mathcal{H}(\pi) = -\int_\mathcal{X} \pi(x)\log\pi(x)d\mu(x)$$

for some target function $f : \mathcal{X} \to \mathbb{R}$ and some temperature parameter $\lambda > 0$. Its closed-form solution is essentially a Boltzmann (softmax) distribution $\pi_{\text{soft}}^f(x) \propto \exp\left(\lambda^{-1}f(x)\right)$ (see Appendix B.3.1), but it is generally much more challenging to estimate the continuous partition factor $\int_\mathcal{X} \exp(\lambda^{-1}f(x))d\mu(x)$ than the normalization factor $\sum_{x \in \mathcal{X}} \exp(\lambda^{-1}f(x))$ in the discrete sample space. Entropy-regularized RL leverages this property and introduces softmax policies $\pi_{\text{soft}}^Q(\cdot|s) \propto \exp\left(\lambda^{-1}Q(s, \cdot)\right)$ by replacing general $f(\cdot)$ with state-dependent value functions $Q(s, \cdot)$, and admits the following protocol to iterate both value functions and policies

$$Q_{k+1} = \mathcal{T}^{\pi_k}Q_k, \quad \pi_{k+1} \propto \exp(\lambda^{-1}Q_{k+1}(s, \cdot)),$$

which equivalently corresponds to value iteration under softmax Bellman operators (Song et al., 2019; Li et al., 2024)

$$Q_{k+1}(s,a) = R(s,a) + \gamma \mathbb{E}_{s' \sim P(\cdot|s,a)} \left[ Q_k(s', \pi_{\text{soft}}^{Q_k}) \right],$$

where hardmax targets are replaced with softmax surrogates, denoted as a softmax Bellman operator $\mathcal{T}_{\text{soft}}$ such that $Q_{k+1} = \mathcal{T}_{\text{soft}} Q_k$. Previous study (Song et al., 2019) analyzed that for discrete control, the aforementioned iteration enjoys a performance bound grounded on the cardinality $\text{Card}_{\mathcal{A}}$ of the finite action set $\mathcal{A}$, which can be detailed as

$$\limsup_{k \to \infty} \left[ Q^*(s,a) - (\mathcal{T}_{\text{soft}}^k Q_0)(s,a) \right] \lesssim O\left( \text{Card}_{\mathcal{A}} \cdot \max\left\{ \frac{1}{\lambda^{-1}+2}, \frac{2V_{\max}}{1+\exp(\lambda^{-1})} \right\} \right). \quad (1)$$

However, in the context of continuous control, a similar suboptimality guarantee still remains unexamined since $\text{Card}_{\mathcal{A}} \to \infty$, highlighting the need for further investigation.

**Policy Gradient.** The mainstream approach to practically train a softmax actor is via the policy gradient (PG) method (Sutton et al., 1999; Schulman et al., 2017; Agarwal et al., 2021)

$$\nabla_\theta J_\lambda(\theta) = \mathbb{E}_{s \sim B, a \sim \pi_\theta} \left[ \nabla_\theta \log \pi_\theta(a \mid s) Q^\pi(s,a) \right] + \lambda \nabla_\theta \mathcal{H}(\pi_\theta),$$

where $s$ is sampled from a minibatch $B = \{(s,a,r,s')\}$, $\pi_\theta$ denotes parametric actors, and $Q^\pi$ is exclusively estimated by a critic network. However, this optimization-based perspective may introduce discrepancy if the parameterized policy class is simple, such as Gaussian families (Fujimoto et al., 2018). Moreover, since a complex policy class may preclude efficiency for sample generations and density estimations, directly applying diffusion models may hinder accurate estimation for the gradients, limiting the applicability of generative PG methods (Ajay et al., 2022; Wang et al., 2024).

**Langevin Dynamics.** In lieu of policy optimizations, instantly sampling from softmax distributions can be an promising alternative, which bypasses the challenge for finding a suitable policy parameterization. Determined by a temperature $\lambda$ and an energy function $E(x)$, the Langevin dynamics (Roberts & Tweedie, 1996) defines a stochastic process

$$dX_t = \frac{1}{2\lambda} \nabla_x E(X_t) dt + dB_t, \quad x_{t+1} = x_t + \frac{\delta_t}{2\lambda} \nabla_{x_t} E(x_t) + \sqrt{\delta_t} \xi_t$$

where the continuous-time Markov chain is formulated on the left, the Euler-Maruyama discretization is demonstrated on the right, $B_t$ denotes standard Brownian motion, $\{\delta_t\}$ represents step schedules, and $\xi_t \sim \mathcal{N}(0, I)$ are i.i.d. Gaussian perturbations. Under mild conditions, both processes will converge to an identical stationary distribution $\pi(\cdot) \propto \exp\left(\lambda^{-1} E(\cdot)\right)$, which leaves room for us to substitute $E(\cdot)$ with $Q(s, \cdot)$ for actor-free softmax action generations. Notably, the Langevin Actor-Critic (LAC) (Lei et al., 2024) and Q-Score Matching (QSM) (Psenka et al., 2025) also adopts the concept of Langevin dynamics, but contrasts with our work from different theoretical perspectives and practical implementations. A detailed comparison is provided in paragraph 10.

While the vanilla Langevin algorithm can effectively resolve $\mathbb{R}^d$ sampling without domain constraints, continuous control problems generally have a finite-volume action space, making the aforementioned MCMC no longer applicable. A common solution is to adopt clamping tricks, but this is in nature the projected Langevin dynamics, which may encounter boundary stagnation problems and cause severe approximation bias. In Section 4, we will provide a MCMC variant with specular reflection to better overcome this issue.

**SNIS Resample.** The self-normalized importance sampling (SNIS) (Kong et al., 1994; Swaminathan & Joachims, 2015) is another feasible alternative for drawing samples from an un-normalized distribution $p(x) \propto u(x)$. Starting from a proposal $q(\cdot)$ that generates $x_1, \dots, x_m$, the SNIS resampler draws $\hat{x}$ among them according to

$$\omega(\hat{x} = x_i \mid x_1, ..., x_m) = \frac{u(x_i)/q(x_i)}{\sum_{i=1}^{m} u(x_i)/q(x_i)}$$

so that the marginal distribution $\omega(\hat{x})$ approximates the target distribution $p$ (see Appendix B.3.5). SNIS enables reusing the same data for estimating various policy statistics (e.g., the softmax entropy), or approximating energy-based actions. However, the SNIS technique may incur high variance or bias when the proposal is not properly designed (Cardoso et al., 2022), making it ill-suited for more refined continuous sample generation. In our setting, we adopt a uniform proposal due to the lack of prior knowledge, making SNIS a coarse initialization scheme for action generation.

# 3 CONTINUOUS SOFTMAX ANALYSIS

While the theory of softmax policy and value iteration is well established for discrete control, a practical challenge arising in continuous control is that softmax policies fail to track hardmax policies without additional assumptions, particularly since the action space cardinality $\text{Card}_{\mathcal{A}}$ diverges to infinity. Intuitively, the difference between the softmax expectation $\mathbb{E}_{a \sim \pi_{\text{soft}}^f}[f(s,a)]$ and the hardmax value $\max_{a \in \mathcal{A}} f(s,a)$ depends not only on the numerical differences between function values but also on how many actions are near-optimal. To quantify this, we can define a volume function that measures the size of the set of actions within a given error threshold $\epsilon$, by capturing how "spread out" the near optimal region is. The definition can be found at Definition 1 as illustrated in Figure 1, and our measure-based analysis directly parallels the regret-based action ranking scheme developed in discrete settings (Song et al., 2019). As a fundamental setup, our analytical framework can be initiated with the following core definitions and assumptions.

**Definition 1.** Given $f$ and $s'$, the regret function is formulated as $\tau_f(a') = f^* - f(s', a')$, where $f^* = \max_{a' \in \mathcal{A}} f(s', a')$.

**Definition 2.** Given $f$ and $s'$, the volume function is defined as $\text{Vol}_{s'}^f(\epsilon) = \mathscr{L}(\{a', \tau_f(a') \leq \epsilon\})$ where $\mathscr{L}$ is the Lebesgue measure over the action space. Additionally, we denote $\text{Vol}_f(\epsilon) = \min_{s' \sim \mathcal{S}}\left[\text{Vol}_{s'}^f(\epsilon)\right]$ as a uniform lower bound over the state space $\mathcal{S}$.

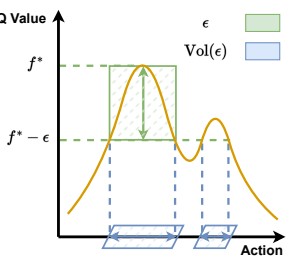 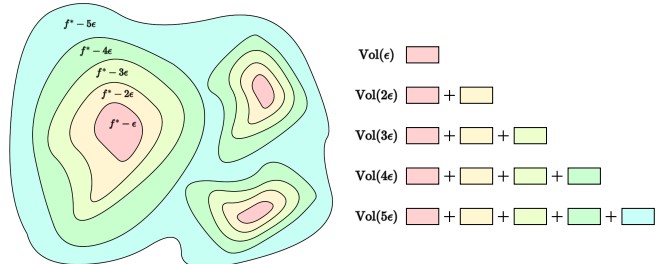

(a) An example 1d volume function plotted in a $(a, f(a))$ fashion.

(b) An example 2d volume function as an diagram with equipotential lines identical in their function values.

Figure 1: A demonstration of volume functions as the integration of colored areas on action space.

**Definition 3.** Given $Q_0$, we assume that the infimum volume function $\text{Vol}(\epsilon) = \inf_{k=0}^{+\infty} \text{Vol}_{Q_k}(\epsilon)$ exists, where $Q_k = \mathcal{T}_{\text{soft}}^k Q_0$.

**Assumption 1.** The action space $\mathcal{A}$ is a Lebesgue-measurable subset with a finite volume $\|\mathcal{A}\| < +\infty$.

**Assumption 2.** Given $Q_0$, there exists a function $g_0(\kappa)$ growing at most polynomially, such that $\forall k \in \mathbb{N}, s' \in \mathcal{S}$, the inequality $\text{Vol}_{s'}^{Q_k}(\kappa\epsilon) \leq g_0(\kappa)\text{Vol}_{s'}^{Q_k}(\epsilon)$ holds for any $\epsilon > 0$, where $Q_k = \mathcal{T}_{\text{soft}}^k Q_0$.

Supported by the above formulations, we are then able to verify a polylog suboptimality bound for the difference between $\mathcal{T}_{\text{soft}}$ and $\mathcal{T}$, as outlined below in Theorem 1.

**Theorem 1.** For a bounded function $Q_0 \in [0, V_{max}]$ and $\forall(s,a) \in \mathcal{S} \times \mathcal{A}$,

$$\liminf_{k \to \infty} Q^*(s,a) - (\mathcal{T}_{\text{soft}}^k)Q_0(s,a) \geq 0 \tag{2}$$

$$\limsup_{k \to \infty} Q^*(s,a) - (\mathcal{T}_{\text{soft}}^k Q_0)(s,a) \lesssim \frac{\gamma}{1-\gamma} O(\lambda \cdot \text{polylog}(\text{Vol}^{-1}(\lambda))) \tag{3}$$

Our proof B.3.3 can be sketched in the following steps. First, we demonstrate that, for any arbitrary function $f$ throughout the iterative updates, the non-negative difference $\mathcal{T}f - \mathcal{T}_{\text{soft}}f$ has an upper bound $O(\lambda \cdot \text{polylog}(\text{Vol}^{-1}(\lambda)))$. This is analyzed by dividing the Lebesgue measure into two parts: the integral on $[0, \kappa\lambda]$ where the majority of the mass is concentrated, and on $[\kappa\lambda, \infty]$ which can be exponentially bounded by $\kappa\lambda \|\mathcal{A}\| \exp(-\kappa)$. Note that by properly choosing a pivot $\kappa$, the error

bound can provably contract to the polylog error term that we desire. Second, we extend the upper bound through mathematical inductions, where

$$\left(\mathcal{T}^k Q_0\right)(s,a) - \left(\mathcal{T}_{\text{soft}}^k Q_0\right)(s,a) \leq \lambda(g_0(\log(\|\mathcal{A}\| \operatorname{Vol}^{-1}(\lambda))) + e\log(\|\mathcal{A}\| \operatorname{Vol}^{-1}(\lambda))) \sum_{j=1}^{k} \gamma^j.$$

Invoking Eq. 3, we thus complete the final proof for polylog suboptimality error bounds.

## 4 ACTOR-FREE LANGEVIN MCMC AND STATISTICAL ESTIMATION

We have now handled one of the main theoretical challenges for continuous softmax policy approximation, but the partition constant still makes it intractable to estimate the policy statistics (e.g. the policy entropy), and so is to draw action samples from the energy-based policies. In Preliminary 2, we demonstrate that the Langevin dynamics can effectively generate Monte-Carlo samples by ruling out the normalization factor, and SNIS estimator can serve as a coarse distributional approximation when the proposal density is not guaranteed to be optimal. Incorporating both techniques, we will next present how to implement softmax samplers without additional parametric actors in this section.

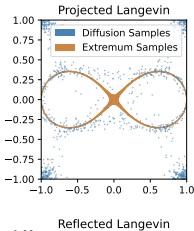

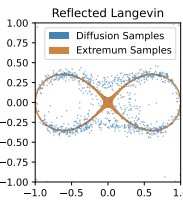

**SNIS Initialization.** To expedite the MCMC sampler, we can reasonably draw uniform samples $a_0^{(0)}, ..., a_0^{(m-1)}$ from the action space, and consecutively resample an initial action $a_0$ with a probability mass function

$$\omega(a_0 = a_0^{(i)} \mid s, a_0^{(0)}, ..., a_0^{(m-1)}) = \frac{\exp(\lambda^{-1} Q(s, a_0^{(i)}))}{\sum_{j=0}^{m-1} \exp(\lambda^{-1} Q(s, a_0^{(j)}))}$$

Figure 2: An illustration for boundary stagnation.

to constitute a principled, network-free initialization protocol, where `logSumExp` techniques are leveraged to ensure numerical stability. Following the analysis in Appendix B.3.5, we will now instantiate the approximation error bound, as demonstrated in Theorem 2.

**Theorem 2.** *Suppose that the proposal $q(\cdot)$ is a uniform distribution over $\mathcal{A}$, and the target generator $p_s(\cdot) \propto \exp(\lambda^{-1} Q(s, \cdot))$. With $m \geq 2$, the total variation (TV) distance between $p_s(\cdot)$ and the marginal density $\omega_m(\cdot \mid s)$ estimated by SNIS procedure, enjoys an error bound*

$$\operatorname{TV}(\omega_m(\cdot \mid s), p_s) \lesssim \frac{\operatorname{Var}_q[W]}{m-1} + m W_{\max} \exp\left(-\frac{m-1}{W_{\max}^2}\right),$$

*where $W$ weights the importance ratio, $\operatorname{Var}_q[W]$ denotes the variance for $W$ under $q$, $W_{\max} = Z^{-1}|\mathcal{A}| \exp(\lambda^{-1} V_{\max})$ represents an upper bound for $W$, and $Z$ is the partition function.*

Nevertheless, we need to reiterate that SNIS is merely a coarse approximation to the target density, since the number of candidate particles $m$ is generally small for efficiency considerations, and the uniform proposal generally does not hold a strong optimality guarantee. For a more fine-grained sample approximation, the discrete-time Langevin MCMC is still necessitated.

**Jitted Score Functions.** In order to run MCMC, a primary task is to access the score function $\nabla E(x)$, which corresponds to $\nabla_a Q_\theta(s, a)$ for continuous

---

**Algorithm 1:** QGLG

**Input** : Observation $s$, potential $Q(\cdot, \cdot)$, temperature $\lambda$, candidate schedules $\Delta_c = \{\delta^{(i)} \in \mathbb{R}^T\}$

**Output:** $a_T$ generated by Langevin MCMC

1 Initialize $a_0$ via the SNIS generator (4) from $Q(s, \cdot)$ and $\lambda$;
2 Pre-sample Gaussian noises $\{\xi_0, ..., \xi_{T-1}\}$;
3 **for** $\delta^{(i)} \in \Delta_c$ **do**
4     **for** $t = 0$ **to** $T - 1$ **do**
5         Set $y_t = \frac{\delta_t^{(i)}}{2\lambda} \nabla_a Q(s, a_t^{(i)}) + \sqrt{\delta_t^{(i)}} \xi_t$ and $a_{t+1} = \mathcal{R}(a_t, y_t)$ (7);
6     Collect terminal sample $a_T^{(i)}$;
7 **return** $\arg\max_i Q(s, a_T^{(i)})$;

---

RL. This can be implemented via jit-compiled gradient functions prior to training, as the network architecture uniquely defines the score function itself, and $\nabla_a Q_\theta(s, a)$ can be instantly determined once $\theta, s$ and $a$ are provided. This approach eliminates the need for additional networks to fit the score function, minimizing the risk of introducing biased score estimates.

**Specular Reflection.** In the preliminary section, we mentioned that projected Langevin dynamics (i.e. Langevin sampler with boundary clipping) may theoretically cause stagnation problems. To visualize this, a counterexample is demonstrated in Figure 2 by exposing the failure of projected Langevin algorithms, where the orange Lemniscate curve matches the point set at which the energy function attains its maximum, and the blue scatter points exhibit i.i.d. samples drawn from a 2-D finite-step Langevin chain, either with boundary projection (the top figure) or specular reflection (the bottom figure) to satisfy domain restrictions. With steps improperly scheduled, the modest number of steps involved in diffusion and the unduly large sizes for initial steps may cause the projected Langevin algorithms to allocate a significant portion of samples at stationary points around boundaries. To this end, we define the operator of specular reflection $\mathcal{R}(x, y)$ as the termination of a free trajectory initiated at $x$ with an initial direction $\frac{y}{\|y\|}$, such that, after free propagation and specular reflections, its total path length amounts to $\|y\|$. Furthermore, a variant with specular reflection can be designed as follows:

$$
\begin{cases}
y_t = \frac{\delta_t}{2\lambda} \nabla_{x_t} E(x_t) + \sqrt{\delta_t}\epsilon_t, & \epsilon_t \sim \mathcal{N}(0, I), \quad \text{\# Compute Langevin Shift} \\
x_{t+1} = \mathcal{R}(x_t, y_t). & \text{\# Law of Reflection}
\end{cases}
$$

This is motivated by the reflected replica exchange stochastic gradient Langevin dynamics (r2SGLD) (Zheng et al., 2024). The basic idea is that the reflected Langevin dynamics will not degenerate to stagnation for steps that may not be well-tuned during training, and the convergence rate for it can be analyzed as follows. We will focus on a simplified case where the sample space is a unit hypercube, as is generally a standard setting for continuous RL, in order to encourage further sophisticated analyses for general bounded point sets.

**Theorem 3.** *With domain $\mathcal{X} = [-1, 1]^d$, (i) the reflection operator $\mathcal{R}$ is an 1-Lipschitz mapping s.t. $\|\mathcal{R}(a, y_1) - \mathcal{R}(b, y_2)\| \leq \|a + y_1 - b - y_2\|$ holds for any $a, b \in \mathcal{X}$ and $y \in \mathbb{R}^d$, and (ii) if the potential function $E(\cdot)$ is m-strongly concave, and $\nabla E$ is $L - Lipschitz$, then the Wasserstein-2 distance, defined as $W_2^2(\mu, \nu) = \inf_{\gamma \in \Pi(\mu, \nu)} \int \|x - y\|^2 d\gamma(x, y)$ for two probability measure $\mu, \nu$ on $\mathbb{R}^d$ with finite second moments (where $\Pi(\mu, \nu)$ is the set of couplings of $\mu, \nu$), is bounded by*

$$
W_2(x^*, x_T) \leq W_2(x^*, x_0) \prod_{t=0}^{T-1} \left(1 - \frac{m\delta_t}{\lambda} + \frac{L^2\delta_t^2}{4\lambda^2}\right),
$$

*where $x^*$ is randomly drawn from the stationary distribution $\mu^*$ of the reflective Langevin dynamics: $\mu^*(dx) \propto \exp(E(x)/\lambda)\mathbf{1}\{x \in \mathcal{X}\}dx$, and $W_2(x^*, x_T)$ is the shorthand for $W_2(\mu^*, \mathscr{L}(x_t))$.*

Furthermore, by substituting $E(\cdot)$ with $Q(s, \cdot)$, the reflective Langevin dynamics can be inherently introduced as a softmax policy approximation conditioned on the state $s$, while enjoying similar convergence guarantees.

**Remark 1.** The exponential rate decay is not always established for a more general class of energy landscapes, if they do not hold strong concave premises. In contrast, as analyzed by Nguyen et al. (2021), potentials devoid of strong concavity may incur an extra term in their upper bound, which is essentially a time-irrelevant discretization bias. This judgment suggests that the final Wasserstein distance may not necessarily vanish even as $T \to \infty$, echoing existing empirical findings (Halder, 2025; Czerwinska, 2025) that under certain circumstances, extending the diffusion chain may yield higher bias compared to shorter-step MCMC.

**Adaptive Step Selection.** Ding et al. (2024) showed that single-chain simulations may lead to degenerated policy performance, which coincides with our observation in early experiments. Though running multiple stochastic chains and selecting the optimal one has become a common choice, it actually biases the generated action from following the target distribution[1]. To address this limitation, we perform a grid search over the step schedule that yields near-optimal actions, with informative SNIS initialization and Gaussian perturbations determined in advance of step selection. Specifically, we adhere to the Q-gradient Langevin generator (QGLG) in algorithm 1 to sample softmax actions from the Q landscapes, while avoiding unnecessary alterations to the underlying softmax–Langevin sampling framework.

---

[1]For example, when the target density is a uniform distribution that generates $x_0, ..., x_{m-1}$, picking an optimal one $\hat{x} = \arg\max_{x \in \{x_0, ..., x_{m-1}\}} f(x)$ is approximately equivalent to finding a $\hat{x} \in \arg\max_{x \in \mathcal{X}} f(x)$ as $m \to \infty$, which is highly biased from the intended $\hat{x} \sim$ uniform.

**Entropy Estimation.**   To facilitate efficient real-time entropy tracking and adaptive temperature control, we can calculate the entropy (see Appendix B.3.4) by

$$\mathcal{H}_Q(\lambda; s) = -\log m + \log \|\mathcal{A}\| + \text{LogSumExp}_{j=1}^m \left( \frac{1}{\lambda} Q(s, a_j) \right) - \frac{1}{\lambda} Q(s, \pi_{\text{soft}}^Q)$$

where $Q(s, \pi_{\text{soft}}^Q)$ can be further estimated via SNIS with uniform proposals. Since the value function exhibits heterogeneous sensitivity to temperature across states, the true differential entropy estimated on real batch data can introduce high variance. This contrasts with prior work such as SAC (Haarnoja et al., 2018) and DACER (Wang et al., 2024), which estimate the statistics via one or multiple Gaussian distributions. The Gaussian-based estimates tend to be inherently more stable than those obtained from arbitrary energy-based distributions, although they may not faithfully reflect the softmax entropy [2].

## 5   EXPERIMENTS AND DISCUSSIONS

**State-Dependent Temperature.**   In practice, the temperature can be state-dependent function $\lambda(s)$ that aligns the magnitude of score functions across different states, without alternating the theory of softmax policy and softmax Q iteration (see Corollary 1). We empirically set this temperature function with z-score normalization

$$\lambda(s) = \lambda_0 \cdot \sqrt{\text{Var}_{a \sim \text{uniform}} [Q(s, a)]}$$

to better mitigate numerical instability, where $\lambda_0$ is a constant that scales the standard deviation.

**Evaluation.**   The deterministic evaluation protocol follows the Q-gradient Langevin generator (Algorithm 1) with specular reflection and step-size selection. The initialization is replaced by a greedy choice

$$a_0 \leftarrow \arg\max_i Q(s, a_0^{(i)}),$$

over uniform candidates, and Gaussian noise is replaced with deterministic zero vectors. Determinism can be ensured with a fixed random seed, and this protocol can thus be seen as a gradient-based DDPG (Lillicrap et al., 2019) action generator with zero-variance perturbations.

---

**Algorithm 2:** DDSQ

**Input** : Temperature $\lambda$, critic params $\theta_1, \theta_2$, target params $\theta_1^-, \theta_2^-$, learning rate $\eta$, soft Polyak rate $\tau$, candidate steps $\Delta_c$
**Output:** $\theta_1, \theta_2, \theta_1^-, \theta_2^-$
1 **for** *each training step* **do**
2    Set twin-Q surrogates
     $Q_\theta(s, a) = \min\{Q_{\theta_1}(s, a), Q_{\theta_2}(s, a)\}$,
     $Q_{\theta^-}(s, a) = \min\{Q_{\theta_1^-}(s, a), Q_{\theta_2^-}(s, a)\}$;
3    Execute environment step through behavior policy (10);
4    Store new data into replay buffer $D$;
5    Sample minibatch $B = \{(s, a, r, s')\} \sim D$;
6    Get target actions via $\pi_\theta = \text{QGLG}(s, Q_\theta, \lambda, \Delta_c)$;
7    Set loss $\mathcal{L}(\theta_i) = \mathbb{E}[(Q_{\theta_i}(s, a) - r - \gamma Q_{\theta^-}(s', \pi_\theta))^2]$;
8    Update critic: $\theta_i \leftarrow \theta_i - \eta \nabla_{\theta_i} \mathcal{L}(\theta_i), i \in \{1, 2\}$;
9    Polyak update: $\theta_i^- \leftarrow \tau \theta_i + (1 - \tau)\theta_i^-, i \in \{1, 2\}$;
10 **return** $\theta_1, \theta_2, \theta_1^-, \theta_2^-$;

---

**Behavior Policy.**   For reconciling exploration with optimality, we thereby adopt a probabilistic combination of Langevin generators and deterministic generators, with likelihoods of $p_e$[3] and $1 - p_e$ respectively. And our training framework is finally presented in Algorithm 2.

**Main Result.**   Our experiments are built upon the publicly available Jax implementation of DACER (Wang et al., 2024), which we selected both for consistency and its capability to model multimodal action distributions. We compare our method against DACER, SAC, and QSM as baselines, and the evaluation is conducted on eight continuous control tasks from the MuJoCo suite: Walker2d-v3, Swimmer-v3, Humanoid-v4, Hopper-v4, InvertedPendulum-v4, HalfCheetah-v4, Pusher-v2, Ant-v4. The hyperparameter settings can be found in Appendix B.2, and the corresponding training curves are presented in Figure 3. The trending performance of our method is comparable to state-of-the-art (SOTA) baselines in most environments, and reaches SOTA performance in several tasks.

---

[2]To this end, we do not directly tune the temperature $\lambda$ itself, and simply set the temperature learning rate equal to zero. However, we do develop an empirical law to calibrate the temperature, as shown in Appendix B.3.4.

[3]We empirically set $p_e = 0.15$ in the experiments.

Compared with DACER, which requires over 20 hours of training, our approach completes training in approximately 10 hours (see Table 1). This demonstrates that our method can effectively balance both optimality and training efficiency.

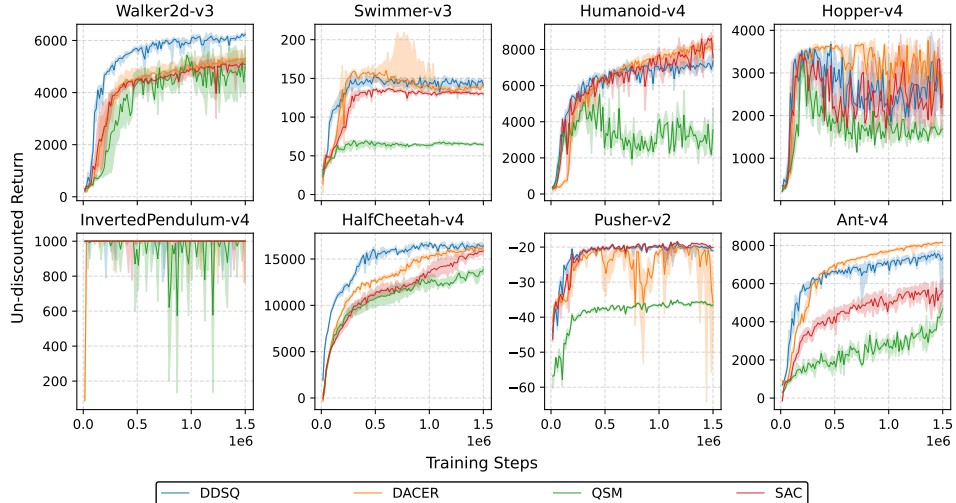

Figure 3: Main results in comparison with the baselines: SAC, QSM, and DACER. The experiments are conducted across 8 MuJuCo benchmarks, aggregating random seeds of 100, 200, 300 and 400 to facilitate a more reliable evaluation for the algorithms involved. The shade surrounding the median curve represents the interval between the 25-th percentile to the 75-th percentile.

| Environment | DDSQ | DACER | QSM | SAC |
|---|---|---|---|---|
| Walker2d-v3 | $10.06 \pm 0.01$ | $22.73 \pm 0.13$ | $6.57 \pm 0.03$ | $1.31 \pm 0.02$ |
| Swimmer-v3 | $10.04 \pm 0.01$ | $22.65 \pm 0.17$ | $6.56 \pm 0.03$ | $1.26 \pm 0.07$ |
| Humanoid-v4 | $11.05 \pm 0.04$ | $23.04 \pm 0.13$ | $7.87 \pm 0.03$ | $1.79 \pm 0.05$ |
| Hopper-v4 | $10.82 \pm 0.05$ | $23.11 \pm 0.25$ | $7.01 \pm 0.17$ | $1.35 \pm 0.13$ |
| InvertedPendulum-v4 | $10.09 \pm 0.05$ | $19.29 \pm 0.25$ | $6.42 \pm 0.01$ | $1.21 \pm 0.04$ |
| HalfCheetah-v4 | $10.10 \pm 0.04$ | $19.62 \pm 0.17$ | $6.43 \pm 0.01$ | $1.24 \pm 0.05$ |
| Pusher-v2 | $9.67 \pm 0.04$ | $22.56 \pm 0.48$ | $6.48 \pm 0.02$ | $1.22 \pm 0.06$ |
| Ant-v4 | $9.68 \pm 0.03$ | $22.50 \pm 0.39$ | $6.51 \pm 0.06$ | $1.51 \pm 0.03$ |

Table 1: A100 hours across environments. Each entry represents a median $\pm$ the interquartile range among the total training time induced by the four random seeds, where each GPU fraction simultaneously parallels 2 running sessions.

**Training-Time Discrepancy.** Discrepancy occurs when the actors are unable to approximate the target softmax distribution given a simple parameterization. This can be empirically verified in Figure 4, where the MCMC sampler in DDSQ faithfully captures multimodal distributions and SAC fails to do so.

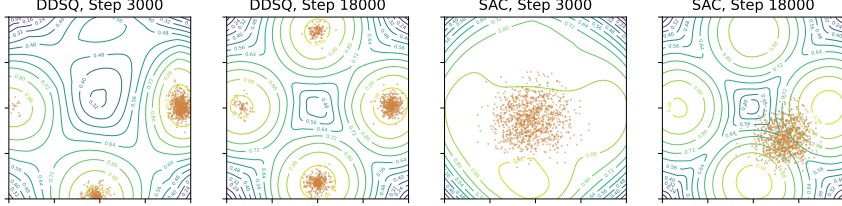

Figure 4: Landscapes of Q functions and actions generated across algorithms at an intermediate training step. The DDSQ here is trained with $\lambda_0 = 0.15$ for a more representative demonstration.

**Test-Time Flexibility.** Compared to standard PG methods that fix the entropy regularization term in the loss and thereby constrain the learned policy to partially align with a certain temperature, our approach offers greater flexibility in controlling the sampling process. As illustrated in Figure

5, even when trained under a relatively small temperature, the converged Q-function can be paired with different temperatures at test time to instantly generate actions with varying trade-offs between diversity and performance. Moreover, when operating at higher temperatures, we also observe that our method achieves higher action diversity than existing baselines such as DACER and SAC.

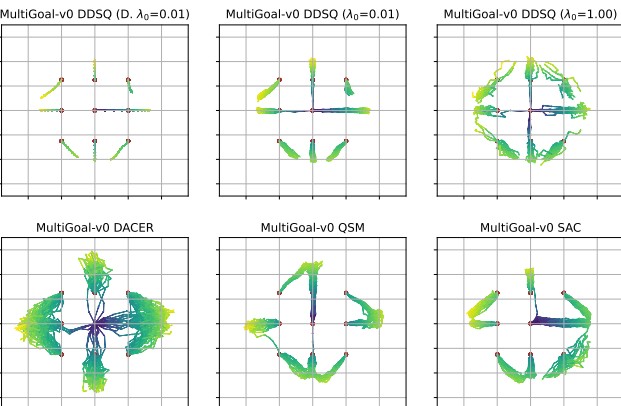

Figure 5: Policies induced by different training algorithms in a 2-D MultiGoal environment (Haarnoja et al., 2017), where the task is to trace targets at $(-5, 0), (5, 0), (0, -5), (0, 5)$. We reset observations at $(2.5 \cdot i, 2.5 \cdot j)$ across different combinations for $i \in \{-1, 0, 1\}$ and $j \in \{-1, 0, 1\}$ at test time, and the first row represents DDSQ policies with diverse test temperatures. The DDSQ here is trained with $\lambda_0 = 0.01$ for a more representative demonstration. Figure titled "D. $\lambda_0 = 0.01$" represents deterministic policy evaluation under scaling factor $\lambda_0 = 0.01$.

**DDSQ, QSM and LAC: A Comparison**    The central ideas of QSM (Psenka et al., 2025) and LAC (Lei et al., 2024) are broadly aligned with Langevin dynamics, yet they differ from our approach either in theoretical starting points, or in practice, by introducing additional approximation errors. Specifically, QSM shows that if a diffusion process parameterized by a score function $s_\theta(a \mid s)$ solves the policy gradient (PG) problem, then the score function must be proportional to the Q-gradient, i.e., $s_\theta(a \mid s) \propto \nabla_a Q(s, a)$. However, this perspective is largely confined to using diffusion models as a PG solver and requires fitting an extra network to approximate the Q-gradient itself, which introduces additional approximation error. In contrast, LAC starts from a different viewpoint, by observing that the solution to constrained policy optimization (CPO) problems (Achiam et al., 2017) naturally takes a softmax form, and hence employs Langevin dynamics to implement sampling. LAC also introduces an auxiliary action network for initialization, but this design inherits limitations of parameterized policy classes. For instance, Gaussian initialization concentrates actions around a single mode that fail to cover multimodal peaks, while diffusion-based initialization incurs higher computational and training costs, potentially leading to suboptimal initialization that requires more diffusion steps. Both QSM and LAC offer limited further insight into the softmax policy itself, and by contrast, our approach provides a refined sampling procedure using SNIS initialization, jitted score functions, specular reflection, and step selection, to enable a more stable Langevin MCMC implementation for softmax policy approximations.

## 6 CONCLUSION AND FUTURE WORK

In this work, we present Deep Decoupled Softmax Q-Learning (DDSQ), a critic-only framework with a deeper understanding of continuous softmax approximation. Our methodology addresses practical limitations of PG optimizations and achieves high-fidelity softmax action generations, and the empirical results on continuous MuJoCo benchmarks demonstrate both strong performance and efficiency. Looking ahead, extending DDSQ to multi-agent scenarios or offline RL would provide opportunities to evaluate its ability to scale to practical applications. We believe that the framework established here can provide a foundation, both theoretically and empirically, for advancing research in high-dimensional policy sampling and softmax-based reinforcement learning.

## ETHICS STATEMENT

We have read and adhered to the ICLR Code of Ethics. Our work does not involve human subjects or sensitive personal data, and all RL environments used are publicly available or properly licensed. We have considered potential societal impacts, including fairness, privacy, and possible misuse, and we believe that our research is conducted responsibly and ethically.

## REPRODUCIBILITY STATEMENT

We are committed to ensuring the reproducibility of our work. Specifically, (i) Our training code builds upon the DACER (Wang et al., 2024) implementation. The code is made available to the reviewers and is also included in the supplementary materials. (ii) All hyperparameters for the algorithms used in our experiments are disclosed in Appendix B.2. (iii) Pseudo-code for our methods is provided in Algorithms 1 and 2.

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

# A  ADDITIONAL EXPERIMENTS

**Fitted Landscape for Different Vol($\lambda$).**    In Figure 6, each panel shows the Q-values across actions for a fixed state. When the optimal $\hat{Q}$ is smooth (high Vol($\lambda$)), the fitted Q-function closely matches $\hat{Q}$. In contrast, when $\hat{Q}$ is sharp (low Vol($\lambda$)), the fitted Q becomes biased, highlighting the difficulty of accurately approximating sharp reward landscapes even under the same temperature $\lambda$.

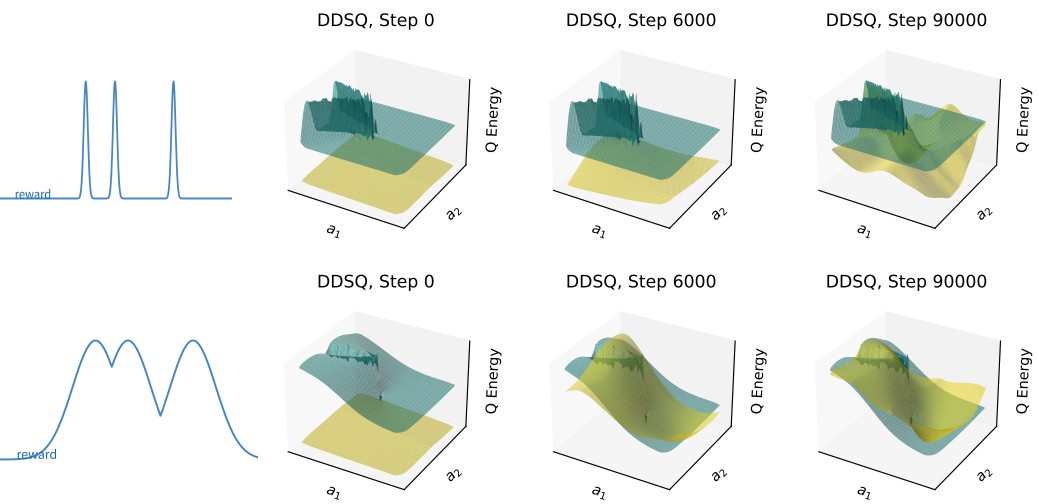

Figure 6: Comparison between learnt and optimal Q-function landscapes.

**Inference Time.**    In Figure 7, we report the inference time for each algorithm. For each method, the inference step was executed consecutively 100 times, and the mean latency was computed over these runs. All single-step action generations were consistently performed on A100 GPUs.

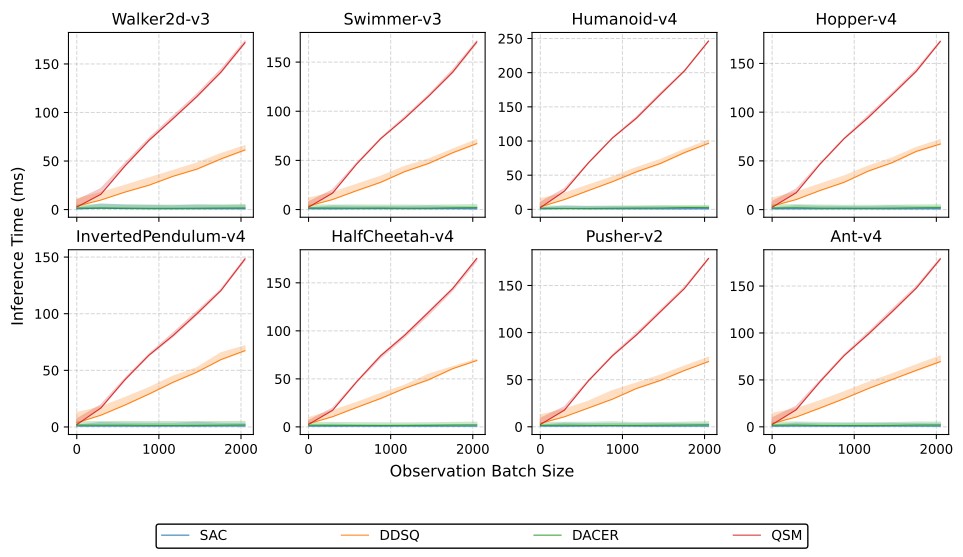

Figure 7:  Inference time of the stochastic behavior policy across algorithms.

**Temperature Selection.**    We show that very small temperatures (0.001) and large temperatures (1) both lead to suboptimal performance—small temperatures correspond to very low differential

entropy, while large temperatures result in imprecise sampling. Intermediate temperatures (0.01 and 0.1) perform well, and we set 0.05 as a compromise in our main experiments.

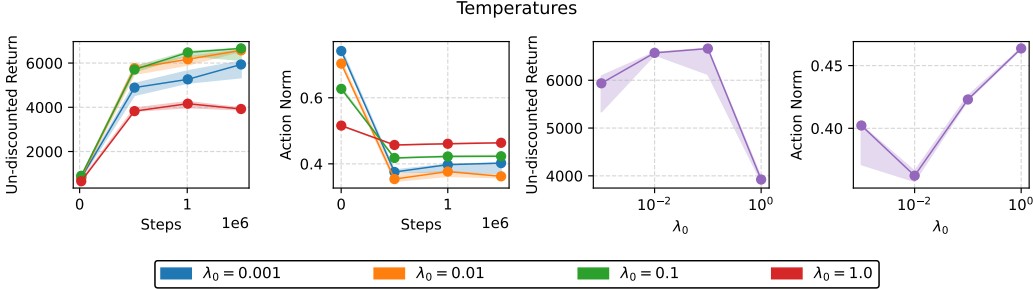

Figure 8: Ablation study for temperature selection.

**Choice of Candidate Steps.** We study the effect of the number of candidate schedules. With only one schedule, sampling degenerates to a single trajectory under linspace[1, 1e-4, 20], yielding the worst performance. Increasing the number of candidate schedules improves performance, which saturates around 8. Considering training cost, we select 4 schedules in the main experiments as a tradeoff.

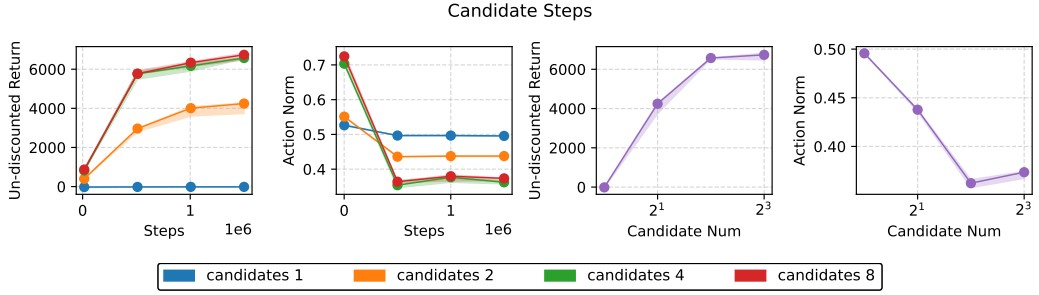

Figure 9: Ablation study for the choice of candidate step schedules.

**MCMC Components.** We compare clip and reflect versions of boundary handling, tracking the average absolute value of each action dimension during training. The clip version leads to more severe stagnation, while reflection mitigates this problem and overall training output is better, consistent with theoretical predictions. In addition, the distribution obtained by SNIS coarse sampling approaches the true softmax distribution as $m \to \infty$. We test $m = 1, 5, 25, 125$ and observe the expected trend, with performance saturating at 125. Notably, when $m = 1$, SNIS degenerates to uniform random sampling, demonstrating the necessity of SNIS for accelerating convergence.

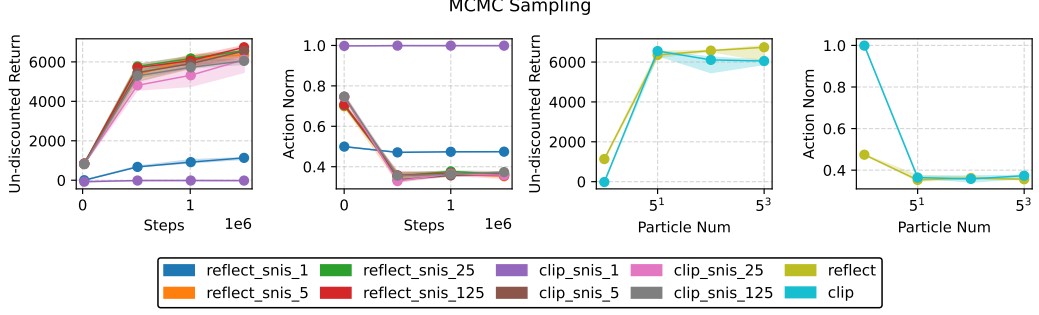

Figure 10: Ablation study for MCMC components.

# B    APPENDIX

## B.1    DISCLOSURE OF LLM USAGE

We use large language models to facilitate our research, and their contributions can be shortlisted as follows. (i) Grammatical refinement. We employ LLM to polish vocabularies, sentences and paragraphs, in order to present the academic paper in a more standard way. (ii) Code development. LLM's are leveraged for debugging and understanding the logic of Jax framework. (iii) Theoretical validation. We leverage LLM to prompt literature research, refer to existing theories, and confirm the feasibility of new theories.

## B.2    HYPERPARAMETERS

To ensure fairness, environment-specific hyperparameters are refrained in our experiments. Instead, each algorithm including DDSQ and other baselines, is evaluated with a unified set of hyperparameters, which is algorithm-specific but sustains environment-agnostic. The concrete settings are detailed in Table 2, 3, 4, 5, and 6, defaulting from the official implementation of DACER (Wang et al., 2024).

| Parameter | Value | Notes |
| --- | --- | --- |
| Vectorized envs | 20 | Number of Environments for Sampling $(s, a, r, s')$ |
| Network architecture | [256, 256, 256] | Commonly adopted neural structures across components |
| Diffusion steps | 20 | Number of steps (if applicable) for diffusion-based methodologies |
| Warmup Steps | 2e5 | Steps involved for warmup data collection |
| Update steps | 1.5e6 | Steps involved for param optimizations |
| Sample steps | 3e6 | Update steps multiplied by vectorized envs |
| Buffer size | 1e6 | Capacity of the replay buffer |
| Batch size | 256 | Batch data for each updtae step |
| Discount factor ($\gamma$) | 0.99 | Standard in MuJoCo tasks |
| Seeds | 100, 200, 300, 400 | Reported with medium $\pm$ interquartile range |

Table 2: Common hyperparameters across algorithms.

| Parameter | Value | Notes |
| --- | --- | --- |
| Reward scale | 1.0 | Reward scaling factor |
| Learning rate | 2e-4 | Adam Optimizer |
| Activation | Mish | Nonlinearity |
| Samples for entropy estimation | 1000 | Uniformly generated for SNIS entropy estimator |
| Samples for SNIS initialization | 100 | Uniform particles for SNIS action initialization |
| Start steps | `logspace(0, -4, 10)` | Candidate start steps |
| End steps | `[1e-4] * 10` | Candidate end steps |
| Initial Temperature $\lambda_0$ | 0.05 | The constant scalar |
| Temperature normalization | `std` | Standard deviation along uniform samples |
| Target entropy | `-act_dim` | Optional, not activated in our experiments |
| Temperature learning rate | 0 | Optional, not activated in our experiments |
| Temperature learning cycle | 100 | Optional, not activated in our experiments |
| $\tau$ | 0.005 | Soft target polyak rate |
| Target update cycle | 1 | $\tau$-exponentially averaged every 1 update steps |

Table 3: Hyperparameters for DDSQ.

| Parameter | Value | Notes |
|---|---|---|
| Reward scale | 0.2 | Reward scaling factor |
| Learning rate | 1e-4 | Adam Optimizer |
| Activation | Mish | Nonlinearity |
| Initial Temperature $\lambda$ | 3.0 | Multiplied by 0.15 for noise injection |
| Samples for entropy estimation | 200 | Generated directly via `get_action` |
| Target entropy | -0.9 * `act_dim` | Estimated via GMM |
| Temperature learning rate | 3e-2 | Optimizing over logarithms of $\lambda$ |
| Temperature learning cycle | 10000 | Update temperatures every 10000 update steps |
| $\tau$ | 0.005 | Soft target polyak rate |
| Target update cycle | 2 | $\tau$-exponentially averaged every 2 update steps |

Table 4: Hyperparameters for DACER.

| Parameter | Value | Notes |
|---|---|---|
| Reward scale | 1.0 | Reward scaling factor |
| Learning rate | 1e-4 | Adam Optimizer |
| Activation | ReLU | Nonlinearity |
| Particles | 64 | Number of i.i.d. Langevin chains for arg-maximum |
| Initial Temperature $\lambda$ | 1.0 | Fixed in Langevin dynamics, not learnable |
| $\tau$ | 0.005 | Soft target polyak rate |
| Target update cycle | 1 | $\tau$-exponentially averaged every 1 update steps |

Table 5: Hyperparameters for QSM.

| Parameter | Value | Notes |
|---|---|---|
| Reward scale | 1.0 | Reward scaling factor |
| Learning rate | 1e-4 | Adam Optimizer |
| Activation | GeLU | Nonlinearity |
| Initial Temperature $\lambda$ | $e$ | The weight of entropy regularization |
| Target entropy | -`act_dim` | Estimated via GMM |
| Temperature learning rate | 3e-4 | Optimizing over logarithms of $\lambda$ |
| Temperature learning cycle | 1 | Update temperatures every 1 update steps |
| $\tau$ | 0.005 | Soft target polyak rate |
| Target update cycle | 1 | $\tau$-exponentially averaged every 1 update steps |

Table 6: Hyperparameters for SAC.

### B.3 PROOFS

#### B.3.1 SOLUTION TO ENTROPY-REGULARIZED OPTIMIZATION

**1. Discrete Case.**

$$\arg\max_\pi \mathbb{E}_{x\sim\pi(\cdot)}[f(x)] + \lambda\mathcal{H}(\pi(\cdot)) = \arg\max_\pi \underbrace{\mathbb{E}_{x\sim\pi(\cdot)}\left[\frac{f(x)}{\lambda}\right] + \mathcal{H}(\pi(\cdot))}_{\text{Linear Scaling}} - \underbrace{\log\sum_{x\in\mathcal{X}}\exp\left(\frac{f(x)}{\lambda}\right)}_{\text{Irrelevant to }\pi,\text{ denoted as }\log Z_f(\lambda)}$$

$$= \arg\min_\pi D_{KL}\left(\pi \left\| \frac{\exp(\lambda^{-1}f(x))}{Z_f(\lambda)}\right.\right),$$

**2. Continuous Case.**

$$\arg\max_\pi \mathbb{E}_{x\sim\pi(\cdot)}[f(x)] + \lambda\mathcal{H}(\pi(\cdot)) = \arg\max_\pi \underbrace{\mathbb{E}_{x\sim\pi(\cdot)}\left[\frac{f(x)}{\lambda}\right] + \mathcal{H}(\pi(\cdot))}_{\text{Linear Scaling}} - \underbrace{\log\int_\mathcal{X}\exp\left(\frac{f(x)}{\lambda}\right)dx}_{\text{Irrelevant to }\pi,\text{ denoted as }\log Z_f(\lambda)}$$

$$= \arg\min_\pi D_{KL}\left(\pi \left\| \frac{\exp(\lambda^{-1}f(x))}{Z_f(\lambda)}\right.\right),$$

both of which induce a softmax solution $\pi_{\text{soft}} \propto \exp(\lambda^{-1}f(x))$, given that the Kullback-Leibler (KL) Divergence attains zero when and only when

$$\pi(x) = \frac{\exp(\lambda^{-1}f(x))}{Z_f(\lambda)}.$$

#### B.3.2 PROOF OF THEOREM 3.

*Proof.* We first prove that, the specular reflection operator $\mathcal{R}$ is a 1-Lipschitz mapping. For a hypercube $\mathcal{X} = \prod_{i=1}^d [L_i, R_i]$, note that for each reflected process along dimension $i$,

$$\mathcal{R}_i(x, a) - L_i = \min_{k_i\in\mathbb{Z}}\|2k_i(R_i - L_i) - x_i - a_i\|,$$

it follows that

$$\|\mathcal{R}_i(x, a) - \mathcal{R}_i(y, b)\| = \left\|\min_{k_i^{(1)}\in\mathbb{Z}}\left\|2k_i^{(1)}(R_i - L_i) - x_i - a_i\right\| - \min_{k_i^{(2)}\in\mathbb{Z}}\left\|2k_i^{(2)}(R_i - L_i) - y_i - b_i\right\|\right\|.$$

Without loss of generality, we assume that

$$\min_{k_i^{(1)}\in\mathbb{Z}}\left\|2k_i^{(1)}(R_i - L_i) - x_i - a_i\right\| \geq \min_{k_i^{(2)}\in\mathbb{Z}}\left\|2k_i^{(2)}(R_i - L_i) - y_i - b_i\right\|$$

and we can further bound the distance by

$$\|\mathcal{R}_i(x, a) - \mathcal{R}_i(y, b)\| = \left\|2k_{i,*}^{(1)}(R_i - L_i) - x_i - a_i\right\| - \left\|2k_{i,*}^{(2)}(R_i - L_i) - y_i - b_i\right\|$$

$$\leq \left\|2k_{i,*}^{(2)}(R_i - L_i) - x_i - a_i\right\| - \left\|2k_{i,*}^{(2)}(R_i - L_i) - y_i - b_i\right\|$$

$$\leq \left\|2k_{i,*}^{(2)}(R_i - L_i) - x_i - a_i - \left(2k_{i,*}^{(2)}(R_i - L_i) - y_i - b_i\right)\right\|$$

$$= \|x_i + a_i - y_i - b_i\|,$$

where $k_{i,*}^{(1)}, k_{i,*}^{(2)}$ denote the $\arg\min$'s, and we hence complete the Lipschitzness proof. Consider some $x_t^* \sim \frac{1}{Z}\exp\left(\frac{E(x)}{\lambda}\right)$ from the stationary distribution induced by the reflective Langevin MCMC, along with an arbitrary $x_t \in \mathcal{X}$. By drawing a $\epsilon_t \sim \mathcal{N}(0, I)$ for both reflected processes,

we obtain that

$$
\begin{aligned}
\left\|x_{t+1}^* - x_{t+1}\right\|^2 &= \left\|\mathcal{R}(x_t^*, y_t^*) - \mathcal{R}(x_t, y_t)\right\|^2 \leq \left\|x_t^* + y_t^* - x_t - y_t\right\|^2 \\
&= \left\|x_t^* - x_t\right\|^2 + 2(x_t^* - x_t)^T (y_t^* - y_t) + \left\|y_t^* - y_t\right\|^2 \\
&= \left\|x_t^* - x_t\right\|^2 + \frac{\delta_t}{\lambda}(x_t^* - x_t)^T (\nabla E(x_t^*) - \nabla E(x_t)) + \frac{\delta_t^2}{4\lambda^2}\|\nabla E(x_t^*) - \nabla E(x_t)\|^2 \\
&\leq \left\|x_t^* - x_t\right\|^2 - \frac{m\delta_t}{\lambda}\left\|x_t^* - x_t\right\|^2 + \frac{L^2\delta_t^2}{4\lambda^2}\left\|x_t^* - x_t\right\|^2 \\
&\leq \left(1 - \frac{m\delta_t}{\lambda} + \frac{L^2\delta_t^2}{4\lambda^2}\right) \left\|x_t^* - x_t\right\|^2,
\end{aligned}
$$

With $\mu_t^* \in \arg\min_{\mu(\cdot,\cdot) \in \Pi(\pi_{\text{soft}}, p_t)} \mathbb{E}_\mu \left[\left\|x_{t+1}^* - x_{t+1}\right\|^2\right]$, it follows that

$$
\begin{aligned}
W_2(x_{t+1}^*, x_{t+1}) = \mathbb{E}_{\mu_{t+1}^*}\left[\left\|x_{t+1}^* - x_{t+1}\right\|^2\right] &\leq \mathbb{E}_{\mu_t^*}\left[\left\|x_{t+1}^* - x_{t+1}\right\|^2\right] \\
&\leq \left(1 - \frac{m\delta_t}{\lambda} + \frac{L^2\delta_t^2}{4\lambda^2}\right) \mathbb{E}_{\mu_t^*}\left[\left\|x_t^* - x_t\right\|^2\right] \\
&\leq \left(1 - \frac{m\delta_t}{\lambda} + \frac{L^2\delta_t^2}{4\lambda^2}\right) W_2(x_t^*, x_t),
\end{aligned}
$$

which leads to a bounded solution

$$
W_2(x^*, x_T) \leq W_2(x^*, x_0) \prod_{t=0}^{T-1}\left(1 - \frac{m\delta_t}{\lambda} + \frac{L^2\delta_t^2}{4\lambda^2}\right). \tag{4}
$$

$\square$

### B.3.3 PROOF OF THEOREM 1

*Proof.* We first prove a single-step bound and then extend it by induction.

**Notation and preliminaries.** For a fixed state $s'$ and function $f$, define

$$
\tau_f(s', a') := f(s', \pi_f^*) - f(s', a'),
$$

where $\pi_f^*(s') = \arg\max_{a' \in \mathcal{A}} f(s', a')$. For $\epsilon \geq 0$ let $\text{Vol}_{s'}^f(\epsilon)$ denote the volume (measure) of the set $\{a' : \tau_f(s', a') \leq \epsilon\}$. We assume Assumption 2 which guarantees a comparison

$$
\text{Vol}_{s'}^f(\kappa\lambda) \leq g_0(\kappa)\,\text{Vol}_{s'}^f(\lambda),
$$

for $\kappa \geq 1$ and some function $g_0(\cdot)$ (as stated in the main text). Also write $\|\mathcal{A}\|$ for the total measure of the action space.

**Step 1: Single-step bound.** For any $f \in [0, V_{\max}]$ we have

$$
\begin{aligned}
(\mathcal{T}f)(s, a) - (\mathcal{T}_{\text{soft}}f)(s, a) &= \gamma\, \mathbb{E}_{s' \sim P(\cdot|s,a)}\left[f(s', \pi_f^*) - f(s', \pi_{\text{soft}}^f)\right] \\
&= \gamma\, \mathbb{E}_{s' \sim P(\cdot|s,a)}\left[\frac{\int_{a'} \tau_f(s', a')\, e^{-\tau_f(s',a')/\lambda}\, da'}{\int_{a'} e^{-\tau_f(s',a')/\lambda}\, da'}\right] \\
&= \gamma\, \mathbb{E}_{s'}\left[\frac{\int_0^\infty \epsilon\, e^{-\epsilon/\lambda}\, d\text{Vol}_{s'}^f(\epsilon)}{\int_0^\infty e^{-\epsilon/\lambda}\, d\text{Vol}_{s'}^f(\epsilon)}\right].
\end{aligned}
$$

Let $g(\epsilon) := \epsilon e^{-\epsilon/\lambda}$. Note $g$ attains its maximum at $\epsilon = \lambda$ with $g(\lambda) = \lambda/e$, and $g$ is decreasing on $[\lambda, \infty)$. Fix $\kappa \geq 1$. Split the numerator:

$$
\int_0^\infty \epsilon e^{-\epsilon/\lambda}\, d\text{Vol}_{s'}^f(\epsilon) = \int_0^{\kappa\lambda} g(\epsilon)\, d\text{Vol}_{s'}^f(\epsilon) + \int_{\kappa\lambda}^\infty g(\epsilon)\, d\text{Vol}_{s'}^f(\epsilon).
$$

Using $g(\epsilon) \le \lambda/e$ on $[0, \kappa\lambda]$ and $g(\epsilon) \le g(\kappa\lambda) = \kappa\lambda e^{-\kappa}$ on $[\kappa\lambda, \infty)$, we get

$$\int_0^\infty \epsilon e^{-\epsilon/\lambda}\, d\mathrm{Vol}_{s'}^f(\epsilon) \le \frac{\lambda}{e}\, \mathrm{Vol}_{s'}^f(\kappa\lambda) + \kappa\lambda e^{-\kappa}\big(\|\mathcal{A}\| - \mathrm{Vol}_{s'}^f(\kappa\lambda)\big).$$

Applying Assumption 2 $(\mathrm{Vol}_{s'}^f(\kappa\lambda) \le g_0(\kappa)\mathrm{Vol}_{s'}^f(\lambda))$ yields

$$\int_0^\infty \epsilon e^{-\epsilon/\lambda}\, d\mathrm{Vol}_{s'}^f(\epsilon) \le \frac{\lambda}{e} g_0(\kappa)\, \mathrm{Vol}_{s'}^f(\lambda) + \kappa\lambda e^{-\kappa}\, \|\mathcal{A}\|\,.$$

For the denominator we have the lower bound

$$\int_0^\infty e^{-\epsilon/\lambda}\, d\mathrm{Vol}_{s'}^f(\epsilon) \ge \int_0^\lambda e^{-\epsilon/\lambda}\, d\mathrm{Vol}_{s'}^f(\epsilon) \ge \frac{1}{e}\, \mathrm{Vol}_{s'}^f(\lambda).$$

Thus, for the choice $\kappa_f := \log\big(\|\mathcal{A}\|/\mathrm{Vol}_{s'}^f(\lambda)\big)$ (take $\kappa_f \ge 1$, e.g. $\kappa_f = \max\{1, \log(\cdot)\}$ if needed), we obtain

$$(\mathcal{T}f)(s, a) - (\mathcal{T}_{\mathrm{soft}}f)(s, a) \le \gamma\lambda\big(g_0(\kappa_f) + e\kappa_f\big).$$

For conciseness define

$$C(\lambda) := g_0\big(\log(\|\mathcal{A}\|\,\mathrm{Vol}^{-1}(\lambda))\big) + e\log(\|\mathcal{A}\|\,\mathrm{Vol}^{-1}(\lambda)),$$

so the single-step bound reads

$$(\mathcal{T}f)(s, a) - (\mathcal{T}_{\mathrm{soft}}f)(s, a) \le \gamma\lambda\, C(\lambda).$$

Finally, by definition $(\mathcal{T}f)(s, a) \ge (\mathcal{T}_{\mathrm{soft}}f)(s, a)$, hence the difference is nonnegative.

**Step 2: Induction.** We prove by induction that for all $k \ge 1$,

$$0 \le (\mathcal{T}^k Q_0)(s, a) - (\mathcal{T}_{\mathrm{soft}}^k Q_0)(s, a) \le \lambda\, C(\lambda) \sum_{j=1}^k \gamma^j.$$

The base case $k = 1$ is exactly the single-step bound above. Now assume the bound holds for $k - 1$. Then

$$(\mathcal{T}^k Q_0)(s, a) - (\mathcal{T}_{\mathrm{soft}}^k Q_0)(s, a)$$
$$= \big(\mathcal{T}(\mathcal{T}^{k-1}Q_0)\big)(s, a) - \big(\mathcal{T}_{\mathrm{soft}}(\mathcal{T}_{\mathrm{soft}}^{k-1}Q_0)\big)(s, a)$$
$$= \Big[\mathcal{T}(\mathcal{T}^{k-1}Q_0) - \mathcal{T}(\mathcal{T}_{\mathrm{soft}}^{k-1}Q_0)\Big](s, a) + \Big[\mathcal{T}(\mathcal{T}_{\mathrm{soft}}^{k-1}Q_0) - \mathcal{T}_{\mathrm{soft}}(\mathcal{T}_{\mathrm{soft}}^{k-1}Q_0)\Big](s, a).$$

For the first bracket we use the standard contraction property of $\mathcal{T}$:

$$\mathcal{T}f - \mathcal{T}g = \gamma\, \mathbb{E}_{s'}\big[\max_a f(s', a) - \max_a g(s', a)\big] \le \gamma\|f - g\|_\infty,$$

hence by the induction hypothesis the first bracket is at most

$$\gamma \cdot \lambda\, C(\lambda) \sum_{j=1}^{k-1} \gamma^j.$$

The second bracket is a single-step difference with $f = \mathcal{T}_{\mathrm{soft}}^{k-1}Q_0 \in [0, V_{\max}]$, so by Step 1 it is bounded by $\gamma\lambda\, C(\lambda)$. Summing these two bounds yields

$$(\mathcal{T}^k Q_0)(s, a) - (\mathcal{T}_{\mathrm{soft}}^k Q_0)(s, a) \le \lambda\, C(\lambda) \sum_{j=1}^k \gamma^j,$$

completing the induction. Nonnegativity holds similarly at each step.

**Step 3: Limit and conclusion.** Combining the above,

$$0 \le \mathcal{T}^k Q_0 - \mathcal{T}_{\mathrm{soft}}^k Q_0 \le \lambda\, C(\lambda) \sum_{j=1}^k \gamma^j = \lambda\, C(\lambda)\frac{\gamma(1 - \gamma^k)}{1 - \gamma}.$$

Taking $k \to \infty$ and using $\mathcal{T}^k Q_0 \to Q^*$ yields the desired bound in Theorem 1. $\square$

While Theorem 1 establishes a polylogarithmic suboptimality guarantee for softmax Q-iteration under a *fixed temperature parameter* $\lambda$, in practice it is often advantageous to allow the temperature to vary across states. The rationale is that different states exhibits distinct action-value landscapes: state with large value gaps benefit from smaller temperatures to sharpen exploitation, whereas flatter landscapes call for larger temperatures to promote adequate exploration.

A natural way to capture this heterogeneity is to adopt a *state-dependent temperature schedule* $\lambda(s)$ (Schulman et al., 2015; Nachum et al., 2017). In particular, a typical choice is the *z-score normalization*:

$$\lambda(s) = \sqrt{\mathrm{Var}_{a\sim\mathrm{uniform}}[Q(s,a)]},$$

which dynamically rescales the local action-value range and thereby enhances algorithmic stability across iterations. Building the same volume-based analysis as in Theorem 1, we can extend the analysis to obtain the following corollary for the state-dependent case.

**Corollary 1.** *Suppose the temperature parameter is chosen as a state-dependent function $\lambda : \mathcal{S} \to \mathbb{R}^+$, and the value function is initialized as $0 \leq Q_0(s,a) \leq V_{\max}$. Then for all $(s,a) \in \mathcal{S} \times \mathcal{A}$, the softmax Q-iteration satisfies the following bounds:*

$$\liminf_{k\to\infty}(Q^* - \mathcal{T}_{\mathrm{soft}}^k Q_0)(s,a) \geq 0, \tag{5}$$

*and*

$$\limsup_{k\to\infty}(Q^* - \mathcal{T}_{\mathrm{soft}}^k Q_0)(s,a) \leq \frac{\gamma}{1-\gamma}\mathbb{E}_{s'\sim d_{s,a}^{\pi_g}}\left[\lambda(s') \cdot \mathrm{polylog}\left(\mathrm{Vol}_{s'}^{-1}(\lambda(s'))\right)\right], \tag{6}$$

*where $Vol_{s'}(\lambda(s')) = \inf_k Vol_{s'}^{Q_k}(\lambda(s'))$, and $d_{s,a}^{\pi_g}$ is the normalized discounted occupancy measure starting from $(s,a)$ under the non-stationary greedy policy $\pi_g$ induced at each iteration. In particular, at round $k$, $\pi_g$ chooses action as*

$$a_k = \pi_g(s_k) = \arg\max_{a\in\mathcal{A}}(\mathcal{T}^{k-1}Q_0 - \mathcal{T}_{\mathrm{soft}}^{k-1}Q_0)(s_k,a). \tag{7}$$

*Moreover, if the state-dependent temperature is bounded such that $\lambda_{\min} \leq \lambda(s) \leq \lambda_{\max}$ for all $s \in \mathcal{S}$, we obtain the uniform upper bound*

$$\limsup_{k\to\infty}(Q^* - \mathcal{T}_{\mathrm{soft}}^k Q_0)(s,a) \leq \frac{\gamma}{1-\gamma}\lambda_{\max} \cdot \mathrm{polylog}\left(\mathrm{Vol}^{-1}(\lambda_{\min})\right). \tag{8}$$

The proof of Corollary 1 follows the same procedure as that of Theorem 1, namely establishing a one-step bound and then applying recursion, except that here the one-step bound is given by

$$(\mathcal{T}f - \mathcal{T}_{\mathrm{soft}}f)(s,a) \leq \gamma\mathbb{E}_{s'\sim P(\cdot|s,a)}\left[\lambda(s') \cdot \mathrm{polylog}\left(\frac{\|\mathcal{A}\|}{\mathrm{Vol}_f^{s'}(\lambda(s'))}\right)\right], \tag{9}$$

from which the greedy policy $\pi_g$ naturally arises when we try to telescope over equation (47).

### B.3.4 DIFFERENTIAL ENTROPY ESTIMATOR

In our continuous-action setting, we estimate the differential entropy of the softmax policy $\pi_{\mathrm{soft}}^Q$ at a given state $s$ as follows. The exact entropy is defined by

$$\mathcal{H}\left(\pi_{\mathrm{soft}}^Q(\cdot \mid s)\right) = \int_{\mathcal{A}} \pi_{\mathrm{soft}}^Q(a \mid s) \log \frac{1}{\pi_{\mathrm{soft}}^Q(a \mid s)} \, da.$$

By substituting the softmax form

$$\pi_{\mathrm{soft}}^Q(a \mid s) \propto \exp\left(\frac{Q(s,a)}{\lambda}\right)$$

and using

$$Z(s,\lambda) = \int_{\mathcal{A}} \exp\left(\frac{Q(s,a)}{\lambda}\right) da,$$

we can rewrite

$$\mathcal{H}\big(\pi_{\text{soft}}^Q(\cdot \mid s)\big) = \log Z(s, \lambda) - \frac{1}{\lambda} Q(s, \pi_{\text{soft}}^Q)$$

$$= \log \mathbb{E}_{a \sim U(\mathcal{A})} \Big[ \|\mathcal{A}\| \cdot \exp \big(\frac{Q(s,a)}{\lambda}\big) \Big] - \frac{1}{\lambda} Q(s, \pi_{\text{soft}}^Q),$$

where $Q(s, \pi_{\text{soft}}^Q)$ is the expected value under the softmax policy. Since the integral over $\mathcal{A}$ is generally intractable, we approximate it using Monte Carlo sampling. Let $a_1, \ldots, a_m \sim U(\mathcal{A})$ be i.i.d. uniform samples. Then the entropy estimator at state $s$ is

$$\hat{\mathcal{H}}_Q(\lambda; s) = -\log m + \text{LogSumExp}_{j=1}^m \Big( \log \|\mathcal{A}\| + \frac{1}{\lambda} Q(s, a_j) \Big) - \frac{1}{\lambda} Q(s, \pi_{\text{soft}}^Q),$$

and its expectation over states defines the overall entropy estimate

$$\mathcal{H}_Q(\lambda) = \mathbb{E}_{s \sim \nu} \big[ \hat{\mathcal{H}}_Q(\lambda; s) \big].$$

To stabilize the temperature update, we also compute the standard error of the estimator,

$$\sigma_Q(\lambda) = \sqrt{\text{Var}_{s \sim \nu} \big( \hat{\mathcal{H}}_Q(\lambda; s) \big)},$$

and update the temperature $\lambda$ via

$$\lambda \leftarrow \lambda - \eta_\lambda \cdot \frac{\mathcal{H}_Q(\lambda) - \bar{\mathcal{H}}}{\max\{1, \sigma_Q(\lambda)\}},$$

where $\bar{\mathcal{H}}$ is the target entropy. The standard error $\sigma_Q(\lambda)$ effectively scales the update to prevent overly aggressive changes when the variance is large.

This estimator is a simple extension of the self-normalized importance sampling (SNIS) approach, where the numerator approximates the log-partition function via the LogSumExp trick and the denominator corrects for the normalization by the sampled softmax policy.

### B.3.5 ANALYSIS OF SELF-NORMALIZED IMPORTANCE SAMPLING

Under specific circumstances, we may wish to draw samples from some target distribution $p$, whereas only a proposal $q$ is given available for practical sample generation. The *self-normalized importance sampling* (SNIS) algorithm (Kong et al., 1994; Swaminathan & Joachims, 2015; Kuzborskij et al., 2021) characterizes the following generative protocol:

1. Draw samples from the proposal distribution $q$:

$$x_1, x_2, \ldots, x_m \overset{i.i.d.}{\sim} q,$$

2. Calculate the weights for importance sampling:

$$W_i = \frac{p(x_i)}{q(x_i)}, \quad \text{where } i \in [m],$$

3. Execute resampling procedure in accordance with the aforementioned weight functions. Output $\tilde{x} = x_I$, where the index $I \in [m]$ is randomly selected with probability

$$\Pr(I = i \mid x_{1:m}) = \frac{W_i}{\sum_{j=1}^m W_j}.$$

In terms of energy-based sample generation, our target distribution $p(x) \propto \exp(f(x))$ corresponds to a softmax distribution with respect to a potential function $f(x)$, and the proposal generator $q(x) = \frac{1}{|\Omega|} \mathbf{1}_{x \in \Omega}$ is specified as a uniform measure, where the normalizing factor over the sample space $\Omega$ is denoted as $Z = \int_\Omega \exp(f(x)) dx$. The analysis will consider a universal class of $p$ and $q$. For our specific choice $p(x) \propto \exp(f(x))$, $q(x) = \frac{1}{|\Omega|} \mathbf{1}_{x \in \Omega}$, the guarantee is shown in Theorem 2. Our theory begins with the following primary result that establishes the asymptotic property of the SNIS estimator, as demonstrated in Lemma 1.

**Lemma 1.** *For a target density $p(x)$ and a proposal distribution $q(x)$ with their weight function being $W(x) = \frac{p(x)}{q(x)}$. If:*

*1. (Non-Negative Support) $\forall x \in \Omega$ such that $p(x) > 0$, the proposal density $q(x) > 0$,*

*2. (Weight Boundness) $\exists W_{\max} > 0$ such that $\forall x \in \Omega$, $0 \leq W(x) \leq W_{\max}$,*

*then $\forall x \in \Omega$, the density $\omega_m(x; x_{1:m})$ estimated by the SNIS generator will converge to the target density*

$$\lim_{m \to \infty} \omega_m(x; x_{1:m}) = p(x),$$

*as $m \to \infty$.*

*Proof.* Since $\omega_m(x; x_{1:m})$ is the probability density function of the output $\tilde{X}$, by definition,

$$\omega_m(x; x_{1:m}) = \sum_{i=1}^{m} \int \cdots \int \frac{W_i}{\sum_{j=1}^{m} W_j} \delta(x - X_i) \prod_{k=1}^{m} q(X_k) dX_1 \cdots dX_m,$$

where $\delta$ is the dirac delta function. Given the i.i.d. property for each $X_i$, we are then able to expand the above target by leveraging symmetry of the random variables, as analyzed accordingly:

$$\omega_m(x; x_{1:m}) = m \int \cdots \int_{x_{2:m}} \frac{W_1}{\sum_{j=1}^{m} W_j} \delta(x - x_1) \prod_{k=1}^{m} q(x_k) dx_1 \cdots dx_m$$

$$= m \int \cdots \int_{x_{2:m}} \frac{\frac{p(x)}{q(x)}}{\frac{p(x)}{q(x)} + \sum_{j=2}^{m} \frac{p(x_j)}{q(x_j)}} q(x) \prod_{k=2}^{m} q(x_k) dx_2 \cdots dx_m$$

$$= m \cdot p(x) \cdot \mathbb{E}_{x_{2:m} \sim q} \left[ \frac{1}{\frac{p(x)}{q(x)} + \sum_{j=2}^{m} \frac{p(x_j)}{q(x_j)}} \right]$$

$$= m \cdot p(x) \cdot \mathbb{E}_{x_{2:m} \sim q} \left[ \frac{1}{W(x) + \sum_{j=2}^{m} W_j} \right]. \tag{$\star$}$$

By the law of large numbers, the following asymptotic guarantee can be yielded as $m \to \infty$:

$$\frac{1}{m-1} \sum_{j=2}^{m} W_j \to \mathbb{E}_{x \sim q}[W(x)] = \int q(x) \frac{p(x)}{q(x)} dx = \int p(x) dx = 1.$$

Incorporating this property, we can ultimately validate the consistency of the SNIS estimator by

$$\omega_m(x; x_{1:m}) \to m \cdot p(x) \cdot \frac{1}{W(x) + (m-1)} \to p(x),$$

where the bounded weight function $W(x)$ is negligible in comparison to $m - 1$. $\qquad \square$

The following establishes a finite-sample, non-asymptotic analysis for the SNIS estimator, confirming the rate depends on both the inherent bias and the concentration deviation.

**Lemma 2.** *Under the assumptions of lemma 1, let $\omega_m(x; x_{1:m})$ denote the output density of the self-normalized importance sampling (SNIS) procedure using $m \geq 2$ samples. Then, for any point $x \in \Omega$ such that $p(x) > 0$, the pointwise bias of the SNIS density estimator is bounded by*

$$|\omega_m(x; x_{1:m}) - p(x)| \leq 2p(x) \left( \underbrace{\frac{|1 - W(x)| + \mathrm{Var}_q[W]}{m-1}}_{\text{polynomial term (bias)}} + \underbrace{\frac{2m(1 + W_{\max})}{W(x)} \exp\left( -\frac{m-1}{2W_{\max}^2} \right)}_{\text{exponential term (concentration)}} \right),$$

*where $W(x) = p(x)/q(x)$, $\mathrm{Var}_q[W]$ is the variance of the importance weights.*

*Proof.* By the expression $(\star)$ of $\omega_m(x; x_{1:m})$, we know that

$$\omega_m(x; x_{1:m}) - p(x) = \mathbb{E}_{x_{2:m} \sim q}\left[ p(x) \cdot \left( \frac{m}{W(x) + S_{m-1}} - 1 \right) \right],$$

where $S_{m-1} = \sum_{j=2}^m W_j$. Let $B(x; x_{2:m}) = p(x) \cdot \left( \frac{m}{W(x) + S_{m-1}} - 1 \right)$ denote the random error inside the expectation. Out goal is to bound $|\mathbb{E}_{x_{2:m} \sim q}[B(x; x_{2:m})]|$.

**Step 1.** Define a high-probability "good event" $\mathcal{E}$ where $S_{m-1}$ is close to its mean.

Since $\mathbb{E}_{x \sim q}[W(x)] = 1$, which means the expectation of the sum is $\mathbb{E}_q[S_{m-1}] = m - 1$, we can define the "good event" $\mathcal{E}$ as the set of outcomes where $S_{m-1}$ does not deviate from its mean by more than a factor of $1/2$:

$$\mathcal{E} = \left\{ |S_{m-1} - (m-1)| \leq \frac{m-1}{2} \right\}.$$

By Hoeffding's inequality, for a sum $S_{m-1}$ of $m - 1$ independent random variables bounded in $[0, W_{\max}]$, we have

$$\Pr\left( |S_{m-1} - \mathbb{E}_q[S_{m-1}]| \geq t \right) \leq 2 \exp\left( -\frac{2t^2}{(m-1)W_{\max}^2} \right).$$

We set $t = (m-1)/2$ to find the probability of the bad event $\mathcal{E}^c$:

$$\Pr(\mathcal{E}^c) \leq 2 \exp\left( -\frac{2((m-1)/2)^2}{(m-1)W_{\max}^2} \right) = 2 \exp\left( -\frac{m-1}{2W_{\max}^2} \right),$$

which decays exponentially in $m$.

**Step 2.** Bounding the bias of the bad event $\mathcal{E}^c$.

We decompose the total bias using the law of total expectation:

$$|\mathbb{E}_{x_{2:m} \sim q}[B(x; x_{2:m})]| \leq |\mathbb{E}_{x_{2:m} \sim q}[B(x; x_{2:m})\mathbf{1}_{\mathcal{E}}]| + |\mathbb{E}_{x_{2:m} \sim q}[B(x; x_{2:m})\mathbf{1}_{\mathcal{E}^c}]|.$$

The second term can be bounded by the supremum of the error multiplied by the probability of the event:

$$|\mathbb{E}_{x_{2:m} \sim q}[B(x; x_{2:m})\mathbf{1}_{\mathcal{E}^c}]| \leq \sup_{x_{2:m} \in \mathcal{E}^c} |B(x; x_{2:m})| \cdot \Pr(\mathcal{E}^c).$$

To bound the error $|B(x; x_{2:m})|$, we need a lower bound on its denominator. Since $S_{m-1} \geq 0$, we have $W(x) + S_{m-1} \geq W(x)$. The numerator is $|m - W(x) - S_{m-1}| \leq m + W(x) + S_{m-1}$. Hence,

$$|B(x; x_{2:m})| = p(x)\frac{|m - W(x) - S_{m-1}|}{W(x) + S_{m-1}} \leq p(x)\frac{m(1 + W_{\max})}{W(x)}.$$

Therefore the contribution from the bad event is thus

$$|\mathbb{E}_{x_{2:m} \sim q}[B(x; x_{2:m})\mathbf{1}_{\mathcal{E}^c}]| \leq p(x)\frac{m(1 + W_{\max})}{W(x)} \cdot 2 \exp\left( -\frac{m-1}{2W_{\max}^2} \right).$$

**Step 3.** Bounding the bias on the good event $\mathcal{E}$.

Our objective is to bound the term $|\mathbb{E}_{x_{2:m}}[B(x; x_{2:m})\mathbf{1}_{\mathcal{E}}]|$. From the definitions, this is equal to $p(x)|\mathbb{E}_q[T \cdot \mathbf{1}_{\mathcal{E}}]|$, where

$$T = \frac{-\delta_S + (1 - W(x))}{W(x) + (m-1) + \delta_S} = \frac{-\delta_S + (1 - W(x))}{a + \delta_S},$$

where $\delta_S = S_{m-1} - (m-1)$ and $a = W(x) + (m-1)$. The expression for $T$ can be rewritten as

$$T = \frac{1 - W(x)}{a + \delta_S} - \frac{\delta_S}{a + \delta_S}$$
$$= \frac{1 - W(x)}{a + \delta_S} - \delta_S\left( \frac{1}{a} - \frac{\delta_S}{a(a + \delta_S)} \right)$$
$$= \frac{1 - W(x)}{a + \delta_S} - \frac{\delta_S}{a} + \frac{\delta_S^2}{a(a + \delta_S)}$$

We now bound the expectation of each of these three terms when multiplied by the indicator $\mathbf{1}_{\mathcal{E}}$. By the triangle inequality,

$$|\mathbb{E}_q[T \cdot \mathbf{1}_{\mathcal{E}}]| \leq \underbrace{\left|\mathbb{E}\left[\frac{1 - W(x)}{a + \delta_S}\mathbf{1}_{\mathcal{E}}\right]\right|}_{(A)} + \underbrace{\left|\mathbb{E}\left[-\frac{\delta_S}{a}\mathbf{1}_{\mathcal{E}}\right]\right|}_{(B)} + \underbrace{\left|\mathbb{E}\left[\frac{\delta_S^2}{a(a + \delta_S)}\mathbf{1}_{\mathcal{E}}\right]\right|}_{(C)}.$$

We first bound the term (A). On the event $\mathcal{E}$, we have $\Delta_S = |S_{m-1} - (m-1)| \leq (m-1)/2$, which implies $S_{m-1} \geq (m-1)/2$. Therefore, the denominator $a + \delta_S = W(x) + S_{m-1}$ is strictly positive and bounded below by $W(x) + (m-1)/2$. Hence,

$$\begin{aligned}
(A) = \left|\mathbb{E}\left[\frac{1 - W(x)}{a + \delta_S}\mathbf{1}_{\mathcal{E}}\right]\right| &\leq \mathbb{E}\left[\frac{|1 - W(x)|}{|a + \delta_S|}\mathbf{1}_{\mathcal{E}}\right] \\
&\leq \mathbb{E}\left[\frac{|1 - W(x)|}{W(x) + (m-1)/2}\mathbf{1}_{\mathcal{E}}\right] \\
&\leq \frac{|1 - W(x)|}{W(x) + (m-1)/2} \\
&\leq \frac{2|1 - W(x)|}{m-1}
\end{aligned}$$

.

We then bound the term (B). Since $\mathbb{E}[\delta_S] = 0$, we have $\mathbb{E}[\delta_S \mathbf{1}_{\mathcal{E}}] = -\mathbb{E}[\delta_S \mathbf{1}_{\mathcal{E}^c}]$. Therefore,

$$\begin{aligned}
|\mathbb{E}[\delta_S \mathbf{1}_{\mathcal{E}}]| = |\mathbb{E}[\delta_S \mathbf{1}_{\mathcal{E}^c}]| &\leq \mathbb{E}[|\delta_S|\mathbf{1}_{\mathcal{E}^c}] \\
&\leq \sup_{x_{2:m} \in \mathcal{E}^c} |\delta_S| \cdot \Pr(\mathcal{E}^c) \\
&\leq (m-1)W_{\max} \cdot \Pr(\mathcal{E}^c),
\end{aligned}$$

where the last step follows because $|S_{m-1} - (m-1)| \leq (m-1)W_{\max}$ is a universal bound. Hence the second term can be bounded by

$$(B) \leq \frac{1}{a}|\mathbb{E}[\delta_S \mathbf{1}_{\mathcal{E}}]| \leq \frac{(m-1)W_{\max}}{W(x) + (m-1)} \cdot 2\exp\left(-\frac{m-1}{2W_{\max}^2}\right),$$

which is exponentially small in $m$.

Finally we bound the term (C). On $\mathcal{E}$, the denominator $a(a + \delta_S)$ is positive and bounded below by $(W(x) + m - 1)(W(x) + (m-1)/2)$. This indicates that

$$\begin{aligned}
\left|\mathbb{E}\left[\frac{\delta_S^2}{a(a + \delta_S)}\mathbf{1}_{\mathcal{E}}\right]\right| &\leq \mathbb{E}\left[\frac{\delta_S^2}{|a(a + \delta_S)|}\mathbf{1}_{\mathcal{E}}\right] \\
&\leq \frac{\mathbb{E}[\delta_S^2 \mathbf{1}_{\mathcal{E}}]}{(W(x) + m - 1)(W(x) + (m-1)/2)} \\
&\leq \frac{\mathbb{E}[\delta_S^2]}{(W(x) + m - 1)(m-1)/2}.
\end{aligned}$$

Since $\mathbb{E}[\delta_S^2] = \mathrm{Var}[S_{m-1}] = (m-1)\mathrm{Var}_{X \sim q}[W(X)]$ by the i.i.d. samples, this becomes

$$(C) \leq \frac{(m-1)\mathrm{Var}_q[W]}{(W(x) + m - 1)(m-1)/2} = \frac{2\mathrm{Var}_q[W]}{W(x) + m - 1} \leq \frac{2\mathrm{Var}_q[W]}{m-1}.$$

Combining these three bounds, we get the bound for the bias on the good event $\mathcal{E}$:

$$|\mathbb{E}_{x_{2:m} \sim q}[B(x; x_{2:m})\mathbf{1}_{\mathcal{E}}]| \leq 2p(x)\left(\frac{|1 - W(x)| + \mathrm{Var}_q[W]}{m-1} + \frac{(m-1)W_{\max}}{W(x) + m - 1}\exp\left(-\frac{m-1}{2W_{\max}^2}\right)\right).$$

**Step 4.** Combining the bounds.

We assemble the final bound on the total bias, which is given by

$$|\omega_m(x; x_{1:m}) - p(x)| = |\mathbb{E}_{x_{2:m}}[B(x; x_{2:m})]| \le |\mathbb{E}_{x_{2:m}}[B(x; x_{2:m})\mathbf{1}_\mathcal{E}]| + |\mathbb{E}_{x_{2:m}}[B(x; x_{2:m})\mathbf{1}_{\mathcal{E}^c}]|.$$

From step 2, we have the bound for the bad event:

$$|\mathbb{E}_{x_{2:m}}[B(x; x_{2:m})\mathbf{1}_{\mathcal{E}^c}]| \le 2p(x)\frac{m(1 + W_{\max})}{W(x)} \exp\left(-\frac{m-1}{2W_{\max}^2}\right).$$

From step 3, we have the bound for the good event:

$$|\mathbb{E}_{x_{2:m}}[B(x; x_{2:m})\mathbf{1}_\mathcal{E}]| \le 2p(x)\left(\frac{|1 - W(x)| + \text{Var}_q[W]}{m-1} + \frac{(m-1)W_{\max}}{W(x) + m - 1}\exp\left(-\frac{m-1}{2W_{\max}^2}\right)\right).$$

Therefore, combining all these together, we get the guarantee in proposition 2:

$$|\omega_m(x; x_{1:m}) - p(x)| \le 2p(x)\left(\underbrace{\frac{|1 - W(x)| + \text{Var}_q[W]}{m-1}}_{\text{polynomial term (bias)}} + \underbrace{\frac{2m(1 + W_{\max})}{W(x)}\exp\left(-\frac{m-1}{2W_{\max}^2}\right)}_{\text{exponential term (concentration)}}\right).$$

$\square$

The structure of the bound in Lemma 2 is highly informative, as it decomposes the total error into two distinct components with different rates of convergence. The dominant component is a polynomial term of order $\mathcal{O}(1/m)$, which represents the intrinsic bias of the estimator and dictates its overall convergence rate. This bias is in turn governed by two key factors: a *local mismatch term*, $|1 - W(x)|$, which captures the inaccuracy at the specific point of evaluation, and a *global mismatch term*, $\text{Var}_q[W]$, which quantifies the overall discrepancy between the proposal and target distributions. The second component of the bound is an exponential term of order $\mathcal{O}(m \cdot e^{-cm})$ that accounts for the risk of a concentration failure, which is common in machine learning literature.

While Lemma 2 characterizes the point-wise bound, it also implies a global measure of distributional error. The following lemma extends the analysis by bounding the total variation (TV) distance between the estimated density $p_m$ and the target density $p$.

**Lemma 3.** *Under the assumptions of Lemma 1, the total variation distance between the estimated density $p_m$ from the SNIS procedure with $m \ge 2$ samples and the target density $p$ is bounded by:*

$$\text{TV}(p_m\|p) \le \frac{\mathbb{E}_p[|1 - W|] + \text{Var}_q[W]}{m-1} + 2m(1 + W_{\max})\exp\left(-\frac{m-1}{2W_{\max}^2}\right).$$

*Proof.* Followed by the definition of the total variation distance, we have

$$\text{TV}(p_m\|p) = \frac{1}{2}\int_\Omega |\omega_m(x; x_{1:m}) - p(x)|dx$$

$$\le \frac{1}{2}\int_\Omega 2p(x)\left(\frac{|1 - W(x)| + \text{Var}_q[W]}{m-1} + \frac{2m(1 + W_{\max})}{W(x)}\exp\left(-\frac{m-1}{2W_{\max}^2}\right)\right)dx$$

$$= \frac{1}{m-1}\left(\mathbb{E}_p[|1 - W|] + \text{Var}_q[W]\right) + 2m(1 + W_{\max})\exp\left(-\frac{m-1}{2W_{\max}^2}\right),$$

where we uses that $\int_\Omega p(x)dx = \int_\Omega q(x)dx = 1$. $\square$

