# OpenReview forum: "Softmax for Continuous Actions: Optimality, MCMC Sampling, and Actor-Free Control"
_ICLR.cc/2026/Conference — Submitted to ICLR 2026_

### Official Review · Reviewer_KH86 · 2025-10-19

**Soundness:** 2
**Presentation:** 2
**Contribution:** 2
**Rating:** 2
**Confidence:** 3

**Summary:**

This paper introduces Deep Decoupled Softmax Q-Learning (DDSQ), an actor-free Maximum Entropy Reinforcement Learning algorithm that attempts to address limitations observed in existing actor-critic methods.

Specifically, the authors highlight that conventional actor architectures often induce overly simplistic action distributions, leading to a mismatch between the learned policy and the true softmax distribution. Furthermore, while more expressive generative architectures can capture complex policies more accurately, they tend to introduce training instabilities and significant computational overhead, especially in the context of maintaining and tuning two complex neural networks (actor and critic), as they introduce extra handcrafted loss functions.

To overcome these challenges, instead of training an actor that approximates the optimal softmax policy, DDSQ leverages Langevin Dynamics to directly sample from it, which it augments with self-normalized importance sampling for particle initialization, specular reflection to prevent boundary stagnation, and a grid-search over step-size schedules to obtain near-optimal action samples. Moreover, the authors provide convergence results for continuous softmax value iteration.

**Strengths:**

- Establishes polylogarithmic suboptimality bound between softmax and hardmax value iteration for continuous actions.

- Though it does not always outperform the showcased baselines, DDSQ balances is significantly faster than the usual best performing one (DACER).

**Weaknesses:**

1. L15: At this level, it is unclear what is meant by continuous softmax approximating hardmax.

2. L46-47 seem to imply that the authors are the first to use a softmax policy over continuous actions, which is inaccurate.

3. The "Premilinaries" section does not contain a paragraph on the actor-critic framework.

4. The softmax distribution is referred to as energy-based (L159) without prior definition.

5. L220 does not define $\tau^k$.

6. The motivation behind establishing the bound to the hardmax policy is not clear. This contribution needs to be better stated.

7. Specular reflection is not defined.

8. The authors state that SNIS expedites convergence, but they do not explain how it does so. Differently, they acknowledge that it is subject to high variance and that it is only used as a "coarse initialization scheme". These two statements seem contradictory.

9. Unclear Langevin-incurred backpropagation computational cost during the critic learning phase.

10. Missing references to similar actor-free work: Messaoud et al. introduced S2AC[1], an SVGD-based actor with a tractable density that is used to estimate the entropy of the SVGD-induced policy. Similarly to LD, the induced distribution is a softmax over Q-values. Differently, the entropy can be approximated in closed form. Also, the authors propose a particle truncation heuristic for divergence control is very similar to SNIS initialization in the case of DDSQ, insofar as it eliminates particles that are not close to the target. S2AC also learns the initial distribution which significantly improves convergence to the target.

[1] Messaoud, S., Mokeddem, B., Xue, Z., Pang, L., An, B., Chen, H., & Chawla, S. S2AC: Energy-Based Reinforcement Learning with Stein Soft Actor Critic. ICLR 2024.

**Questions:**

1. What is $\mu(x)$ in L155.

2. L61: How are these candidate step schedules defined in DDSQ? They are not properly introduced in section 4.

3. Why is MCMC not applicable in constrained action spaces in L146?

4. What exactly is boundary stagnation?

5. What is the formula for specular reflection?

6. When learning the critic, are you backproping through the Langevin Dynamics steps? If yes, how expensive is this?

7. How many LD steps are used in the experiments?

---

> ### Author Response · Authors · 2025-11-27
> **Rebuttal from Authors**
>
> We thank the reviewer for the comments and respond to the questions below.
>
> ---
>
> **Hard max v.s. Softmax, and Motivation**
>
> The motivation for studying how well the hard max can track the softmax is to examine whether entropy regularization is always justified in continuous MDPs. In essence, we aim to establish a metric that **quantifies the intrinsic “hardness'' of entropy-regularized RL.** While multimodal peaks may indeed arise in the optimal soft value function $\hat{Q}$ induced by a continuous MDP, it is also possible for learned critics to exhibit a clear mismatch from the true target $Q^{\*}$. In such cases, the resulting bias arises from the softmax-based value iteration itself, rather than from other RL-theoretical factors such as data coverage constants or function class realizability & Bellman completeness. This hardness is directly influenced by the regularization parameter $\lambda$, and our analysis shows that the volume function $\mathrm{Vol}(\lambda)$ plays a central role in characterizing this hardness---an aspect that has largely been overlooked in prior work on continuous action spaces. In addition, we provide an illustrative example in the appendix, showing that the fitted Q landscape closely approximates the optimal $Q^{\*}$ when $Q^{\*}$ is smooth (corresponding to a high $\mathrm{Vol}(\lambda)$), whereas it becomes highly biased when $Q^*$ is sharp, even under the same temperature $\lambda$.
>
> ---
>
> **The First to Use Continuous Softmax?**
>
> We respectfully disagree with this accusation, as our writing never indicates that we are the first to propose continuous softmax (this would be a ridiculous claim). Sampling from a softmax distribution has long been recognized as a **fundamental challenge** in entropy-regularized reinforcement learning, particularly in continuous action spaces. A number of prior works have been cited in the preliminaries, including [1], [2], [3], [4], and [5], whose contributions have been clearly acknowledged and appreciated in our writing.
>
> [1] Tuomas Haarnoja, Haoran Tang, Pieter Abbeel, and Sergey Levine. Reinforcement learning with deep energy-based policies, 2017. URL https://arxiv.org/abs/1702.08165.
>
> [2] Tuomas Haarnoja, Aurick Zhou, Pieter Abbeel, and Sergey Levine. Soft actor-critic: Off-policy maximum entropy deep reinforcement learning with a stochastic actor, 2018. URL https://arxiv.org/abs/1801.01290.
>
> [3] Fenghao Lei, Long Yang, Shiting Wen, Zhixiong Huang, Zhiwang Zhang, and Chaoyi Pang.
> Langevin policy for safe reinforcement learning. In Forty-first International Conference on Ma-
> chine Learning, 2024. URL https://openreview.net/forum?id=xgoilgLPGD.
>
> [4] Haitong Ma, Tianyi Chen, Kai Wang, Na Li, and Bo Dai. Soft diffusion actor-critic: Efficient online reinforcement learning for diffusion policy, 2025. URL https://arxiv.org/abs/2502.
> 00361.
>
> [5] Michael Psenka, Alejandro Escontrela, Pieter Abbeel, and Yi Ma. Learning a diffusion model policy from rewards via q-score matching, 2025. URL https://arxiv.org/abs/2312.11752.
>
> ---
>
> **Preliminaries Should Contain Actor-Critic**
>
> We respectfully note that Actor-Critic methods are a **well-established and widely-known** class of reinforcement learning algorithms. Their basic formulation—consisting of an actor that updates the policy and a critic that estimates the value function—is considered common knowledge in the field. For this reason, we focused our preliminaries on concepts directly relevant to our key contributions.
>
> ---
>
> **$\mathcal{T}^k$ Not Defined?**
>
> We would like to clarify that the Bellman operator $\mathcal{T}$ (which is defined on Line 90) and its repeated application $\mathcal{T}^k f = \mathcal{T}(\mathcal{T}^{k-1} f)$ are standard and foundational concepts in RL and dynamic programming, as presented in classical textbooks. In fact, value iteration is often succinctly written as $\mathcal{T}^k f_0$ where $f_0$ is the initialization. Consequently, the concern that “$\mathcal{T}^k$ is not defined’’ most likely arises from a misunderstanding of these well-established conventions, rather than from any ambiguity in our presentation. We hope this clarification resolves the issue.
>
> ---
>
> **Definition of Specular Reflection**
>
> Thank you for raising this point. We have clarified on this point with reviewer xHyJ. It was defined verbally in the “specular reflection” paragraph. Specifically, we are considering a particle at position $x$ that is subjected to an auxiliary motion vector $y$, where $\frac{y}{\Vert y\Vert}$ represents its initial direction of motion and $\Vert y\Vert$ indicates the length of path that the particle is going to travel in total. It is evident that once the initial direction and total travel length are specified, the motion of the particle becomes fully deterministic, with the particle undergoing specular reflections at the boundaries. And **the reflection operator $\mathcal{R}$ maps such $(x,y)$ to its final destination according to this mechanism.**

---

> ### Author Response · Authors · 2025-11-27
>
> **”Contradictory” SNIS Initialization and High Variance?**
>
> We respectfully disagree. Theorem 2 establishes that the distribution obtained via SNIS coarse sampling converges to the true softmax distribution as the number $m$ of actions sampled from uniform proposals approaches infinity. While SNIS initialization enjoys such theoretical guarantees, we can never employ $+\infty$ many candidate particles for realistic computation. When $m<\infty$ is finite, the marginal distribution of $\hat{x}$ is intrinsically (and slightly) biased the true softmax distribution, that is why we call it “coarse” approximation, and a more fine-grained adjustment like the Langevin dynamics is still necessitated especially in high-dimensional action spaces.
> The concept of "high variance,'' however, refers to $\text{Var}_{s}\big[\mathcal{H}(\pi(\cdot|s))\big]$, which is entirely unrelated to SNIS samplers. Our key point is that the actual policy entropy across different states can be **arbitrarily large**, making it challenging to provide accurate mean entropy estimates with a small batch size. While prior works achieve controllable differential entropy estimation by assuming Gaussian policies or GMMs (with a limited number of components), they fail to statistically capture the true differential entropies whose distribution can be far more complex than Gaussian families. From a straightforward derivation, one can see that such entropy can be estimated by L327 using $m$ uniform samples, which is essentially a **closed form, classical integral approximation through uniform expectation**. It appears that the reviewer has substantial conceptual misunderstandings regarding the distinction between these two unrelated terms, and their interpretation on our methodologies is fundamentally incorrect, which we would like to clarify.
>
> ---
>
> **Cost of Langevin Dynamics**
>
> If you are referring to the computational cost of running the full MCMC chain, the inference speed is now demonstrated in the appendix, and we would like to thank you for your favorable suggestions. In addition, the training times reported in the main text also reflect the computational cost of our MCMC-based, critic-only training algorithm.
>
> ---
>
> **Comparison with S^2AC: Entropy and SNIS Initialization?**
>
> We respectfully note that we are unable to fully understand the reviewer’s criticism, particularly the part following "differently.'' Our entropy estimation is derived directly and solely from the definition of differential entropy. It does not involve any additional or complicated approximation techniques. The estimator simply computes the integral of the differential entropy by drawing uniform samples over a finite interval---a mathematically straightforward, natural, and fully justified closed-form approximation. Therefore, we do not agree with the reviewer’s characterization that our entropy estimation is "not closed-form'', which we believe is based on a misunderstanding of our method.
>
> Regarding the SNIS initialization, it requires **no additional learning**. Given any candidate $Q$ function (or critic, in the context of deep RL), the initialization produces an approximation that is mathematically well-founded and comes with a clear theoretical guarantee: the discrepancy provably converges to zero as the sample size $m$ increases. This property follows directly from standard results on self-normalized importance sampling.
>
> Additionally, we would like to clarify that S$^2$AC operates by applying a kernel at each iteration to transform a set of particles at step $t$ into a new set at step $t+1$, using a score-function-based weighted aggregation. This mechanism is fundamentally different from---and not directly related to---our initialization procedure. Hence, we believe the reviewer may have conflated these two distinct concepts. We hope this clarification helps resolve the confusion and demonstrates that both our entropy estimator and our SNIS initialization are mathematically sound, simple, and fully justified.

---

> > ### Author Response · Authors · 2025-11-27
> >
> > **Step Schedules, and LD Steps**
> >
> > Please kindly refer to the table of hyperparameters disclosed in our Appendix.
> >
> > ---
> >
> > **L155 $u(x)$ Definition**
> >
> > Please kindly refer to L153 where we define an un-normalized distribution $p(x)\propto u(x)$.
> >
> > ---
> >
> > **Boundary Stagnation, and Why is MCMC “Inapplicable” in L146?**
> >
> > The Langevin MCMC chain defined in the preliminaries section follows the update rule
> > $$
> > x_{t+1} = x_t + \frac{\delta_t}{2\lambda} \nabla_{x_t} E(x_t) + \sqrt{\delta_t}\xi_t,
> > $$
> > from which it is evident that $x_{t+1}$ may fall outside the domain $\mathcal{X}$ whenever $\delta_t$, $\nabla_{x_t} E(x_t)$ or $\xi_t$ is relatively large, and this behavior reflects a standard issue of interest in the field of optimization. While clipping remains the most widely adopted practice in deep RL studies, such operations can lead to a concentration of samples at the boundary—a phenomenon we term “boundary stagnation.” We also provide a clear illustrative example of this effect in Figure L225–241.
> >
> > We hope that our clarification can address part of your concerns, and your detailed suggestions are appreciated.

---

### Official Review · Reviewer_xHyJ · 2025-11-01

**Soundness:** 2
**Presentation:** 2
**Contribution:** 2
**Rating:** 2
**Confidence:** 3

**Summary:**

This paper studies the softmax (Boltzmann or energy-based) policies with respect to a Q function in reinforcement learning. Firstly, the paper provides a suboptimality gap of the stationary point of the softmax operator. Then the paper proposes DDSQ, a new value-based algorithm with MCMC sampling for continuous control. DDSQ uses self-normalized importance sampling (SNIS) to sample the initialization points of Langevin dynamics and specular reflection to adjust the sampling step due to the bounded action space. The paper gives some theoretical analysis of both techniques. Empirically, it shows the proposed algorithm achieves competitive performance in Gym MuJoCo and demonstrates multimodal action distributions in specific cases.

**Strengths:**

The paper has the following strengths:
1. The problem of sampling from the Boltzmann policies of the Q function has been discussed and of interest to the community (e.g., Haarnoja et al., 2017; Messaoud et al., 2024).
2. It provides some useful theoretical analysis. Section 3 discusses a suboptimality gap of the softmax operator, which characterizes how the optimality gap grows with respect to the temperature and the introduced volume function. To my knowledge, such a relationship has not been studied for RL with continuous actions.
3. Overall, it is well written with good figures to explain concepts and phenomena. Figures 1 and 2 illustrate the volume function and stagnation issue of the Langevin dynamics with bounded space very clearly.The mathematical notation is also mostly clear.

Haarnoja, T., Tang, H., Abbeel, P., & Levine, S. (2017). Reinforcement learning with deep energy-based policies. ICML.

Messaoud, S., Mokeddem, B., Xue, Z., Pang, L., An, B., Chen, H., & Chawla, S. (2024). S^2AC: Energy-Based Reinforcement Learning with Stein Soft Actor Critic. ICLR.

**Weaknesses:**

Meanwhile, the paper also has some weaknesses that need to be addressed:
1. Limited empirical investigation.
  - While strong performance has been achieved in MuJoCo environments, it is unclear how the proposed methods work in other continuous control environments (e.g., DeepMind Control Suite, MetaWorld etc.). Considering there are many algorithmic components and hyperparameters, it’s unclear whether the choice of these components and hyperparameters are robust and still work in other domains.
  - In addition, it is unclear if and how much each component impacts learning. A proper ablation study is needed to understand them better.
2. Limited discussion of and comparison to closely relevant works. While the paper compares the DDSQ with a few relevant works, it misses a very closely related work (Messaoud et al., 2024). Messaoud et al. also investigate different sampling methods for Boltzmann policies. The paper should provide a comparison to their method and include their algorithms as baselines.

While theoretical statements in the paper are nice, they are not technically strong and are not the core contributions of the paper. On the other hand, the current empirical investigation is limited, so I tend towards a negative assessment of the paper at this stage.

**Questions:**

Could the authors clarify the below questions?
1. I couldn’t find a clear definition of specular reflection. What does it do specifically?
2. What’s the definition of $W$ in Theorem 2? It’s mentioned to “weigh the importance ratio”, but what is it exactly?
3. Theorem 2 includes the cardinality of the action space in the bound. Does that imply that this result is mostly useful in the discrete case but not the continuous setting?
4. It appears that Theorem 3 doesn’t necessarily imply exponential rate decay. Could the authors provide a discussion on the conditions under which it is such a decay?
5. What’s the impact of mixing the stochastic and deterministic generators? How important and sensitive is this design choice?
6. Similar to 5, what’s the impact of SNIS initialization, specular reflection, and the grid search for the sampling step size schedule?
7. What’s the inference speed of the proposed sampling strategy, compared to something like the Gaussian policy in SAC?

Other minor questions and suggestions with little impact on the rating:
1. Line 200: It might be a good idea to give some intuition about assumption, describing how it restricts the shape of the Q function. For example, what kind of Q functions would be more likely to satisfy the assumption, and when would the assumption be violated.
2. Lines 263 - 269: the discussion of jitted implementation should be put in the experiment section, which reduces interruption to the flow of introducing the core algorithms.
3. Some references (like (4) (7) (10)) are not referencing equations or anything that is properly labeled. It’s better to avoid them to reduce confusion.
4. Footnote 2 and Appendix A.3.4 together are a bit confusing – they contradict with the state-dependent temperature (Line 344). I can only infer how the temperature is set after reading the hyperparameter table. Perhaps add more explanation so that it’s clear what is used.
5. The hyperparameter table in Appendix A.2 does not include $p_e$. Maintaining a complete list of hyperparameters helps improve reproducibility.

---

> ### Author Response · Authors · 2025-11-27
> **Rebuttal from Authors**
>
> We thank the reviewer for the comments and respond to the questions below.
>
> ---
>
> **Comparison to Messaoud et al., 2024**
>
> Due to limited time, we are unable to incorporate this at the moment.  We have prioritized more comprehensive studies, including large-scale ablations and the construction of a visual MDP to illustrate the hardness of softmax value iteration.  We hope to explore additional baseline comparisons in future work.
>
> ---
>
> **Specular reflection**
>
> It was defined verbally in the “specular reflection” paragraph. Specifically, we are considering a particle at position $x$ that is subjected to an auxiliary motion vector $y$, where $\frac{y}{\Vert y\Vert}$ represents its initial direction of motion and $\Vert y\Vert$ indicates the length of path that the particle is going to travel in total. It is evident that once the initial direction and total travel length are specified, the motion of the particle becomes fully deterministic, with the particle undergoing specular reflections at the boundaries. And **the reflection operator $\mathcal{R}$ maps such $(x,y)$ to its final destination according to this mechanism.**
>
> ---
>
> **Definition of $W$**
>
> Please kindly refer to L1190.
>
> ---
>
> **Theorem 2 includes the cardinality of the action space in the bound.**
>
> Our |A| is the volume of the action space (see Assumption 1), not cardinality. We use Card_A (see Eq.(1)) to denote the cardinality to distinguish between the two cases.
>
>
> ---
>
> **Theorem 3 doesn’t necessarily imply exponential rate decay**
>
> Thank you for the comment. Indeed, exponential decay can be obtained by a standard argument.
> Choosing $\delta_t$ such that
> $
> \frac{m \delta_t}{\lambda} \ge \frac{L^2 \delta_t^2}{2 \lambda^2} \quad \text{and} \quad 1 - \frac{m \delta_t}{2\lambda} \ge 0,
> $
> we have
> $
> \prod_{t=0}^{T-1} \Big(1 - \frac{m \delta_t}{\lambda} + \frac{L^2 \delta_t^2}{4\lambda^2}\Big)
> \le \prod_{t=0}^{T-1} \Big(1 - \frac{m \delta_{\min}}{2\lambda}\Big)
> \le \exp\Big(- T \frac{m \delta_{\min}}{2\lambda}\Big),
> $
> where $\delta_{\min} = \min_t \delta_t$. This is a standard technique in convex optimization.
>
> ---
>
> **impact of mixing the stochastic and deterministic generators**
>
> Thank you for highlighting this point. Upon revisiting our experiments, we found that mixing MCMC stochastic policies with deterministic sampling is not strictly necessary, and doing so would compromise the simplicity and elegance of our method. Therefore, we will remove this hybrid behavior and rely solely on the softmax policy itself as the behavior.
>
> ---
>
>
> **impact of SNIS initialization, specular reflection, and the grid search for the sampling step size schedule**
>
> We acknowledge that in the initial submission, we omitted key ablations and comparisons necessary to verify the importance and necessity of the newly introduced components. We have now added these experiments in the appendix, showing the training trends on a grid of size 4, with each data point averaged over three random seeds to estimate error bars. We use the Ant-v4 environment, which effectively differentiates the training performance of different algorithms and hyperparameters.
>
> We provide ablations on the following aspects:
>
> 1. **Temperature control:** We show that very small temperatures (0.001) and large temperatures (1) both lead to suboptimal performance—small temperatures correspond to very low differential entropy, while large temperatures result in imprecise sampling. Intermediate temperatures (0.01 and 0.1) perform well, and we set 0.05 as a compromise in our main experiments.
>
> 2. **Number of candidate particles $m$ for SNIS initialization:** According to Theorem 2, the distribution obtained by SNIS coarse sampling approaches the true softmax distribution as $m \to \infty$. We test $m=1,5,25,125$ and observe the expected trend, with performance saturating at 125. Notably, when $m=1$, SNIS degenerates to uniform random sampling, demonstrating the necessity of SNIS for accelerating convergence.
>
> 3. **Boundary handling (reflection vs clipping):** We compare clip and reflect versions of boundary handling, tracking the average absolute value of each action dimension during training. The clip version leads to more severe stagnation, while reflection mitigates this problem and overall training output is better, consistent with theoretical predictions.
>
> 4. **Candidate schedules:** We study the effect of the number of candidate schedules. With only one schedule, sampling degenerates to a single trajectory under linspace[1, 1e-4, 20], yielding the worst performance. Increasing the number of candidate schedules improves performance, which saturates around 8. Considering training cost, we select 4 schedules in the main experiments as a tradeoff.
>
> 5. **Other components (e.g., diffusion steps, double Q-learning):** These are consistent across all baselines, so additional ablations are unnecessary.
>
> ---
>
> **Inference Speed**
>
> The inference speed is now demonstrated in the appendix. Thanks for your suggestions.

---

### Official Review · Reviewer_riCg · 2025-11-01

**Soundness:** 3
**Presentation:** 3
**Contribution:** 3
**Rating:** 4
**Confidence:** 3

**Summary:**

Sampling in proportion to $\exp Q(s,\cdot)$ is one of the greatest challenges of SAC as we are seemingly unable to parametrize the inverse CDF as an MLP. This limits the representation capabilities of the policy. The author propose an MCMC approach for sampling directly from $\exp Q(s, \cdot)$ with a warm starting approach. This approach differs from existing literature (except the one case mentioned in the questions) in that they do not learn a forward diffusion policy. This simplifies and accelerates training greatly. Experimental results show that the method is competitive in MuJoCO domains while also being able to sample from multi-modal distributions.

**Strengths:**

The paper solves a real problem. It does so elegantly while not introducing yet another thing to learn. This to me is the best part of this work and I think the community will find it interesting.

This work is interesting on the theoretical side as well. This work extends Song et al.'s result on the suboptimality of softmax policies to (bounded) continuous actions. The results are clear and to the best of my knowledge correct. Similarly, the MCMC (with initialization) algorithm is similarly analyzed.

**Weaknesses:**

The writing is confusing at the beginning, I was expected entropy regularized RL (a la SAC) not the softmax Bellman operator.

The experiments do not include deepmind control nor do they try to ablate the effect of different components like SNIS initialization, reflection correction, automatic temperature tuning, double Q learning, and the added compute requirement. In particular, a comparison with TD3PG and explicitly gradient would have been appreciated.

A non toy example where multi modality is useful would have greatly elevated the paper, an example of this would be ant maze where one of the path is randomly blocked at test time.

**Questions:**

Do you have an intuition on the tradeoffs of using forward vs backward mode differentiation for calculating the gradient of the energy function? Which one do you use.

What happens if Assumption 1 does not hold? I understand that the second integral in theorem 1 becomes unboundable, but do you think that similar results could hold or do you have a counter example?

Is Assumption 2 really reasonable? Is this equivalent to bounded gradients for Q?

In theorem two, can W be defined explicitly? Similarly can R (reflection) be also defined in the main text?

Do you think that it would be relevant to cite and compare with [1] ?


# nits
105 -> what property
can the bar in Figure 3 be made thicker?
866 what does $\mu(\cdot,\cdot)\in\Pi$ mean?


[1] Vineet Jain, Tara Akhound-Sadegh, and Siamak Ravanbakhsh. "Sampling from Energy-based Policies using Diffusion." Reinforcement Learning Journal, vol. 6, 2025, pp. 2291–2307.

---

> ### Author Response · Authors · 2025-11-27
> **Rebuttal from Authors**
>
> We thank the reviewer for the comments and will improve writing accordingly. Below we respond to your questions.
>
> ---
>
> **tradeoffs of using forward vs backward mode differentiation? Which one do you use**
>
> We are unsure what the reviewer means by “forward/backward mode differentiation.” If forward mode refers to adding a network to approximate Q-function gradients and backward mode to computing exact gradients via auto-diff, we note that adding a network increases approximation error and complexity. In our work, we compute exact Q gradients using auto-diff and jitted JAX functions, as described in Section 4 under “jitted score functions.”
>
> ---
>
> **usefulness of multi-modality**
>
> We agree that demonstrating multimodality in complex, non-toy environments is important but challenging, as visualizing or quantifying differences between policies (e.g., in robot locomotion) is not straightforward. While using Ant-Maze is a valid suggestion, it is currently a lower priority due to time constraints. Our focus during the rebuttal is on large-scale ablations and creating a 2D environment to illustrate the hardness of softmax approximation under pathological volume growth. We plan to explore multimodality visualization in these environments in future work.
>
> ---
>
> **Assumption 1**
>
> Assumption 1 virtually holds in nearly all practical scenarios; for example, in robotics the action is often force or torque exerted on a motor or joint, which is always finite. If the action space is infinite, we may need to make additional assumptions on the Q-function such as a fast decay when action gets large, but that will complicate the analyses, and given the lack of real-world motivations we don’t think it’s worth it.
>
> ---
> **Assumption 2 reasonable? Is it equivalent to bounded gradients for Q**
>
> Our goal is to show that as temperature $\lambda$ decreases, the gap between hard max and softmax vanishes, which is the case in discrete softmax. For continuous actions, this will be impossible without some assumption like Assumption 2: imagine a pathological Q-function where the maximum is achieved on a single point or generally a zero-measure set, and all other points in the space have value substantially lower than this maximum (note that the function is not continuous). In this case, however low the temperature, the continuous softmax cannot get arbitrarily close to the hardmax.
>
> In terms of bounded gradients for Q, we believe they are not equivalent; in fact, our assumption should be generally weaker. Indeed, if Q has bounded gradients, it should avoid the aforementioned pathological case, since in that example the Q-function is not continuous, let alone smooth or differentiable. However, our assumption only looks at the volume of the level sets. Imagine the following conceptual experiment: take a nice Q with bounded gradients that satisfy Assumption 2, then swap different regions of its input space in an arbitrary manner. Note that the volume of level sets will be intact in this operation so Assumption 2 will still hold, but the operation can clearly destroy the differentiability of the function. That said, later we do rely on the boundedness of gradient to sample with Langevin dynamics, but that’s for sampling from softmax in a computationally efficient manner. It is not needed in the soft value iteration analysis where the focus is on the gap between softmax and hard max, and the softmax is assumed to be computed exactly.
>
> ---
>
> **Definition of $W$ and $R$**
>
> The definition of $W$ is in L1190; we will introduce it in the main text. For reflection $R$, our analysis is flexible and not restricted to a particular form of reflection, and Theorem 3 holds as long as $R$ satisfies the 1-Lipschitz condition on Line 293. That said, we will introduce concrete examples of $R$ and show that the 1-Lipschitz condition is satisfied. As a simple example, consider the standard mirror reflection against a hyperplane (i.e., the action space is one side of the hyperplane): the condition on Line 293 is an equality if a+y1 and b+y2 are both within the boundary (in which case $R$ does nothing) or both out of the boundary (in which case both get reflected), but can be a strict inequality when only one of them is reflected.
>
> ---
>
> **Comparison to [1]**
>
> We thank the reviewer for the pointer and agree that this work [1] falls within the classical Gaussian diffusion framework, where a continuous-time forward process transforms $x_0 \sim \exp(-E(x))/Z$ into near-pure noise, and the backward SDE recovers samples consistent with the target distribution. The score in Eq.~(4) is elegantly derived and approximated via Monte Carlo samples. But this work further introduces a score network $f_\phi$ to approximate the Monte Carlo score, adding approximation error, and its initialization at $\tau = 1$ does not bring the initial distribution closer to the target---a gap our SNIS initialization addresses. We may cite this work in the final manuscript if space permits.

---

### Official Review · Reviewer_jyj3 · 2025-11-04

**Soundness:** 2
**Presentation:** 1
**Contribution:** 2
**Rating:** 2
**Confidence:** 2

**Summary:**

The paper studies entropy‑regularized optimization and its use in softmax Q‑iteration. It claims a polylogarithmic sub‑optimality gap between hard and soft Bellman operators, analyzes a reflective Langevin sampler on bounded domains, proposes a state‑dependent temperature schedule, and gives a differential‑entropy estimator for continuous actions along with a brief SNIS analysis.

**Strengths:**

The paper tackles a problem that if solved could allow for KL regularization in continuous action MDPs.

**Weaknesses:**

It seems this papers SNIS setup can require exponentially many samples in the effective horizon $H=1/(1-\gamma)$ which for the standard discount factor of $\gamma = 0.99$ would require $m = \exp(100)$. This seems like an argument against such a method since it suffers from the so-called curse of the horizon [1].

In general the paper is hard to follow and would benefit greatly from more care and polish in its presentation.


[1] Liu, Qiang, et al. "Breaking the curse of horizon: Infinite-horizon off-policy estimation." Advances in neural information processing systems 31 (2018).

**Questions:**

1. Is it really reasonable to assume to minimum volume is lower bounded?
2. Can you somehow tune $\lambda$ to avoid the exponential blow up in theorem 2 or is this unavoidable?


some comments

1. Definition 3 seems like an assumption ("we assume" is stated in the definition).
2. Definition 2 please separate the lower bound on the volume as an assumption. Since this is essentially definition 3 (which should be an assumption), you should introduce the notation here.
2. line 35, are -> is
3. line 081, include $\gamma \in [0,1)$.
4. line 143, what is paragraph 10, is it the 10th paragraph in your manuscript (somewhere on page 3 or 4?) or the tenth \paragraph{} wrapper? Maybe call it a remark ?
5. line 131: do you mean policy gradient (not policy optimization) since policy optimization is the process of learning a good policy (fitted $Q$ iteration (DQN) is a policy optimization method)?

---

> ### Author Response · Authors · 2025-11-27
> **Rebuttal from Authors**
>
> We thank the reviewer for their comments. It seems, however, that the reviewer has major misunderstanding on the nature of our algorithm, which we clarify below.
>
> **Curse of Horizon**
>
> We are very familiar with the notion of curse of horizon, such as discussed by Liu et al [1]. The curse of horizon refers to the exponential variance of multi-step importance sampling estimator for off-policy evaluation. See also a clear exposition on the problem in a recent survey on offline RL (Jiang & Xie, 2025, Section 2). The key element in multi-step importance sampling that leads to the curse of horizon is the **cumulative product of per-step importance weights over the horizon**, e.g., $\prod\_{h=1}^H \pi(a\_h | s\_h) / \pi\_b(a\_h |s\_h)$ for behavior policy $\pi\_b$ and target policy $\pi$. While we also use importance sampling, **the usage is restricted within a timestep to approximate the expectation of a hard-to-sample distribution, and our importance weights are not multiplied across time steps.** Concretely, upon receiving a **single** input state $s$, a collection of random particles $a_1,...,a_m$ will be generated from a proposal density $q$;  if we resample a candidate $\hat{x}$ using PMF $\omega(\hat{x}=x_i | x_{1:m})=\frac{u(x_i)/q(x_i)}{\sum_{i=1}^m u(x_i)/q(x_i)}$, it follows that the marginal density for $\hat{x}$ will approach the target distribution $\frac{u(x)}{Z}$ as long as $m$ is sufficiently large. This avoids the calculation of the the partition factor $Z=\int_{\mathcal{X}}u(x)dx$ which is generally intractable. In terms of softmax policies, we specify $u(x)=\exp\left(\frac{Q(s,a)}{\lambda}\right)$ and $q(x)=\text{uniform over action spaces}$, which directly derives the result in Section 4. Therefore, the notion of curse of horizon is completely irrelevant and not applicable to our work.
>
> ---
>
> **Def 2 & 3 as assumptions / Is it really reasonable to assume the minimum volume is lower bounded?**
>
> We agree Definition 3 is actually an assumption and will revise. For Definition 2, while the existence of infimum is technically an assumption (we will also revise), it is not really an assumption in the practical sense. Note that we need infimum because we are taking the minimum value across all iteration numbers $k=0, \ldots, \infty$. In practice, however, no one will run (soft) value iteration to infinite iterations, and we essentially only need to define Vol($\epsilon$) as the minimum over $k=0, \ldots, K$ for $K$ being the maximum iteration number; this way, it will be 100% a definition and not an assumption. That said, to state a standard theoretical results such as Eq.(3), it is standard to consider $k\to \infty$ (otherwise we will have a residual term proportional to $\gamma^K$ if we stop at a finite iteration), so the “assumption” in Definition 2 is merely a technicality to handle the infinity here.
>
> ---
>
> **Can you somehow tune $\lambda$ to avoid the exponential blow-up in Theorem 2?**
>
> As we have clarified earlier, your understanding that we suffer the curse of horizon is incorrect. Moreover, Theorem 2 actually establishes a polynomially-weighted exponential decay with respect to the number of candidate samples $m$, rather than an exponential blowup as you suggested.
>
> ---
>
> **Policy gradient, Not Policy Optimization**
>
> We believe this is a terminology issue. While policy optimization has slightly different connotations in different parts of the literature, our understanding is that it generally refers to methods that find a good policy within a policy class, for which policy gradient and actor critic are special cases. The reviewer seems to associate “policy optimization” with value-based methods such as Fitted-Q Iteration (FQI) or DQN. We believe the former (our usage) is more popular, which is reflected by the names of widely used algorithms such as TRPO (trust region policy optimization) and PPO (proximal policy optimization).
>
>
>
> References
>
> Jiang & Xie. Statistical Science, 2025. Offline Reinforcement Learning in Large State Spaces: Algorithms and Guarantees.

---

### Meta-Review · Area_Chair_y1Lx · 2026-01-05

**Summary:**

**1) Summary**
The paper investigates entropy-regularized optimization and softmax Bellman operators, proposing DDSQ—a value-based continuous-control algorithm that replaces a learned actor with MCMC sampling enhanced by SNIS initialization and specular reflection. It contributes theoretical analysis of softmax suboptimality gaps and convergence, and provides empirical results showing competitive performance in MuJoCo with multimodal policies. While novel in approach, the paper’s presentation, empirical depth, and contextualization within recent related work are limited.

**2) Strengths**

1. Addresses a meaningful challenge in sampling from Boltzmann/softmax policies in continuous action spaces, offering an elegant actor-free alternative that simplifies training.
2. Provides useful theoretical insights, including a polylogarithmic suboptimality gap and analysis of reflective Langevin sampling and SNIS-based initialization.
3. Demonstrates competitive performance in MuJoCo and illustrates multimodal action distributions, showing both practical promise and conceptual clarity in several figures.

**3) Weaknesses**

1. Empirical evaluation is limited in scope (e.g., missing DMControl, MetaWorld) and lacks necessary ablations to understand the contribution of individual components (SNIS, reflection, temperature tuning, etc.).
2. Writing and exposition are frequently unclear, with several undefined terms, missing contextualization, and confusing early sections that hinder readability.
3. The computational overhead of Langevin sampling during critic updates is not clearly quantified, making it difficult to compare efficiency with standard actor-critic methods.
4. The motivation and practical significance of the theoretical bounds (e.g., softmax–hardmax suboptimality) are not well articulated, reducing clarity on how they inform algorithm design.
5. Key methodological choices—such as specular reflection, temperature scheduling, and SNIS initialization—lack justification or comparison to alternatives, leaving uncertainty about robustness and necessity.

**4) Summary of author-reviewer discussion**
The reviewers had a consensus of rejection for this paper, and did not seem to change their mind after the authors' rebuttal.

**Reviewer Concerns:**

The reviewers did not follow up on the authors' rebuttal.

**Reviewer Scores:**

The reviewers did not follow up on the authors' rebuttal. Hence, they have not changed their scores.

---

### Decision · Program_Chairs · 2026-01-26

Reject